# Solar and volcanic forcing of North Atlantic climate inferred from a process-based reconstruction

Jesper Sjolte[1], Christophe Sturm[2], Florian Adolphi[1,3], Bo M. Vinther[4], Martin Werner[5], Gerrit Lohmann[5], and Raimund Muscheler[1]

[1]Department of Geology – Quaternary Science, Lund University, Sölvegatan 12, 223 62, Lund, Sweden.
[2]Department of Geological Sciences, Stockholm University, SE-106 91 Stockholm, Sweden.
[3]Climate and Environmental Physics & Oeschger Centre for Climate Change Research, Physics Institute, University of Bern, Sidlerstrasse 5, CH-3012 Bern, Switzerland
[4]Centre for Ice and Climate, Niels Bohr Institute, University of Copenhagen, Juliane Maries Vej 30, DK-2100 Copenhagen Oe, Denmark.
[5]Alfred Wegener Institute, Helmholtz Centre for Polar and Marine Sciences, Bussestr. 24, 27515 Bremerhaven, Germany.

**Correspondence:** Jesper Sjolte (jesper.sjolte@geol.lu.se)

**Abstract.** The effect of external forcings on atmospheric circulation is debated. Due to the short observational period the analysis of the role of external forcings is hampered, making it difficult to assess the sensitivity of atmospheric circulation to external forcings, as well as persistence of the effects. In observations, the average response to tropical volcanic eruptions is a positive North Atlantic Oscillation (NAO) during the following winter. However, past major tropical eruptions exceeding the magnitude of eruptions during the instrumental era could have had more lasting effects. Decadal NAO variability has been suggested to follow the 11-year solar cycle, and linkages has been made between grand solar minima and negative NAO. However, the solar link to NAO found by modeling studies is not unequivocally supported by reconstructions, and is not consistently present in observations for the 20th century. Here we present a reconstruction of atmospheric winter circulation for the North Atlantic region covering the period 1241-1970 CE. Based on seasonally resolved Greenland ice core records and a 1200-year long simulation with an isotope enabled climate model, we reconstruct sea level pressure and temperature by matching the spatio-temporal variability of the modeled isotopic composition to that of the ice cores. This method allows us to capture the primary (NAO), and secondary mode (Eastern Atlantic pattern) of atmospheric circulation in the North Atlantic region, while, contrary to previous reconstructions, preserving the amplitude of observed year-to-year atmospheric variability. Our results show 5 winters of positive NAO on average following major tropical volcanic eruptions, which is more persistent than previously suggested. In response to decadal minima of solar activity we find a high-pressure anomaly over Northern Europe, while a reinforced opposite response in pressure emerges with a 5-year time lag. On centennial time scales we observe a similar response in circulation as for the 5-year time-lagged response, with a high-pressure anomaly across North America and south of Greenland. This response to solar forcing is correlated to the second mode of atmospheric circulation, the Eastern Atlantic pattern. The response could be due to an increase in blocking frequency, possibly linked to a weakening of the subpolar gyre. The long-term anomalies of temperature during solar minima shows cooling across Greenland, Iceland and Western Europe, resembling the cooling pattern during the Little Ice Age (1450-1850 CE). While our results show significant correlation between solar forcing and the secondary circulation pattern on decadal ($r = 0.29$, $p < 0.01$) and centennial timescales

(r = 0.6, p < 0.01), we find no consistent relationship between solar forcing and NAO. We conclude that solar and volcanic forcing impacts different modes of our reconstructed atmospheric circulation, which can aid to separate the regional effects of forcings and understand the underlying mechanisms.

## 1  Introduction

Climate variability in the North Atlantic region can, to a large extent, be explained by different modes of atmospheric circulation. This is particularly true for the variability during winter. The dominant mode is the NAO, which is a measure of the strength and position of the westerly winds across the North Atlantic. Traditionally, the NAO is defined as the pressure difference between Iceland and the Azores (Walker and Bliss, 1932). Using gridded data, the NAO can be defined as the first Principal Component (PC1) of Sea Level Pressure (SLP) in the North Atlantic region (Hurrell et al., 2003). The second most important pattern, PC2 of SLP, is often referred to as the Eastern Atlantic pattern (Wallace and Gutzler, 1981). Modes of circulation can also be identified using cluster analysis, and the secondary modes include Atlantic Ridge and Trough -types of variability, characterized by mid-Atlantic and Scandinavian blockings, respectively (Hurrell et al., 2003). These different modes for example determine the severity of European winters (Hurrell et al., 2003). In paleoclimate studies climate variability and changes atmospheric circulation are often attributed to external forcing related to solar variability or volcanic eruptions (e.g., Ortega et al., 2015; Wang et al., 2017)). Using reanalysis of weather observations (1871-2008) it has been shown that a positive phase of the NAO occurs in the winters following major tropical volcanic eruptions, while climate models generally fail to reproduce this dynamical response to the forcing (Driscoll et al., 2012; Zambri and Robock, 2016; Swingedouw et al., 2017). For the past millennium, Ortega et al. (2015) found a positive NAO response in the second winter following the 11 largest tropical eruptions in their reconstructed NAO. However, more persistent climate effects of volcanic eruptions were found by Sigl et al. (2015) who inferred cooler European summer temperatures lasting up to 10 years following major tropical eruptions during the past 2,500 years, raising the question if a more persistent impact on winter circulation also can be expected. The observed NAO does not show a consistent correlation to the solar forcing (Gray et al., 2013). However, anomalies in SLP during 1950-2010 exhibit a NAO-like pattern correlated to the 11-year solar cycle (Ineson et al., 2011; Adolphi et al., 2014). It has been hypothesized that solar-induced anomalies in the stratosphere can propagate to the troposphere (Haigh and Blackburn, 2006; Kodera and Kuroda, 2005) possibly synchronizing NAO variability to the 11-year solar cycle (Thieblemont et al., 2015) with a maximum response lagging 2-4 years due to ocean memory effects (Scaife et al., 2013; Gray et al., 2013). Several paleoclimate studies have indicated solar influences on climate in the North Atlantic region on centennial time-scales (Bond et al., 2001; Sejrup et al., 2010; Moffa-Sanchez et al., 2014; Adolphi et al., 2014; Jiang et al., 2015), and it has been suggested that the climate conditions during the Little Ice Age was linked to negative NAO forced by low solar activity (Shindell et al., 2001; Swingedouw et al., 2011). However the nature of the solar influence in terms of mechanisms and dynamical response is debated.

Our understanding of past climate dynamics and the impact of external forcings relies heavily on analysis of past changes. Gridded data sets of climate variables based on meteorological observations have been developed for these purposes (Compo et al.,

2011). Such reanalysis data sets are constrained in quality and coverage back in time due to sparse instrumental data, and we rely on climate reconstructions based on proxy data to go beyond the instrumental era. Previously, reconstructions of climate indices (such as e.g. the NAO) have been site-based, with the reconstruction essentially done by extrapolating observed empirical relationships back in time. This approach results in a wide spread of reconstructions of past atmospheric circulation modes (Pinto and Raible, 2012). Recently, efforts have been done to develop reanalysis-type climate reconstructions based on climate proxy data. However, those reconstructions are focused on annual mean data, not taking into account the migration of circulation patterns from summer to winter (Hurrell et al., 2003), and have low skill for atmospheric circulation over the North Atlantic (Hakim et al., 2016; Steiger et al., 2017).

Stable water isotope ratios in ice cores carry quantitative information about past climate (Johnsen and Vinther, 2007). For Greenland ice cores seasonal isotope variability is attainable. In particular, the winter isotope signal has been shown to be highly correlated to atmospheric circulation and temperature (Vinther et al., 2003, 2010). However, the strength of the relation between the main patterns of variability of the ice core isotope signal and the NAO has been suggested to vary in strength (Vinther et al., 2003), indicating that a simple (regression based) relation between the ice core records and NAO bears large uncertainties for reconstruction purposes.

Here we present a climate reconstruction for the North Atlantic region for winter covering 1241-1970 CE, and analyze the impact of solar and volcanic forcing. For our reconstruction we combine a simulation using a coupled atmosphere-ocean model with stable isotope diagnostics embedded in the hydrological cycle with eight seasonally resolved isotope records from Greenland ice cores. We do not calibrate the reconstructed meteorological variables to observations, as we solely rely on matching the modeled isotopic composition to the ice core data. Testing the reconstruction against reanalysis data and observations, the reconstruction has good skill not only for the NAO, but also for secondary circulation modes. We find the average response to major tropical volcanic eruptions to be a positive NAO for the five consecutive winters after eruptions, which is more persistent than previous studies have shown. However, we find no persistent relationship between solar forcing and the NAO. On the other hand, we find a strong impact of solar forcing on the secondary modes of circulation represented by PC2 of reconstructed SLP. We achieve the strongest correspondence between the solar forcing and reconstructed PC2 of SLP with a time lag of 5 years, indicating that an atmosphere-ocean feedback is in play. Taking this time lag into account we find a consistent relationship between PC2 of reconstructed SLP and solar forcing on decadal to centennial time scales.

## 2 Data and Methods

### 2.1 Model simulation

We use the isotope enabled version of the atmosphere-ocean model ECHAM5/MPI-OM (Werner et al., 2016) to simulate the period 800-2000 CE forced by greenhouse gases, volcanic aerosols, total solar irradiance, land-use and orbital forcing (see Table 1). For our study ECHAM5 is run with a T31 spectral resolution ($3.75^o$ x $3.75^o$) with 19 vertical layers, 5 of which are in the stratosphere. Our simulation uses a similar set up as the E1 COSMOS ensemble by Jungclaus et al. (2010), except for an updated solar forcing (see section below) and fully prescribed $CO_2$, where the E1 simulations incorporates a carbon cycle

module. We apply the identical physical general circulation models (GCMs) (ECHAM5/MPI-OM) as the E1 ensemble, as well as similar forcings, and our simulation generally yields a very similar climate as the E1 ensemble. The performance of the atmospheric component of the model used in this study, ECHAM5-wiso, was evaluated for the Arctic region and Antarctica using different configurations of spatial resolution by Werner et al. (2011). For the configuration used in this study (T31) the

model has a warm bias and is not depleted enough in $\delta^{18}$O, however the climatological relation between $\delta^{18}$O and temperature compares well to observations, despite the relatively course resolution. Greenland is represented by 50 grid points in the simulation.

## 2.2    Solar forcing

The solar forcing record employed in this study is based on the solar modulation record inferred from the combined neutron

monitor and tree-ring $^{14}$C data (Muscheler et al., 2016, 2007). In contrast to Schmidt et al. (2011, 2012) it covers the last 2000 years and consistently uses the solar modulation record for the complete period, while in Schmidt et al. (2011, 2012) the $^{14}$C-based record is combined with sunspot-based data for the period after 1850 CE. Therefore, our approach employs an internally self-consistent forcing record for the last 2000 years, and agrees well with the latest recommended solar forcing reconstruction for the past millennium (Jungclaus et al., 2017). The 11-year solar cycle is based on the neutron monitor and $^{14}$C data for the

last 500 years where the underlying data has a sufficiently high temporal resolution. For the period before 1500 CE an artificial 10.5-year cycle was added and the phasing was adjusted to allow for a smooth connection to the subsequent start of the data-based solar cycle. The amplitude of the solar cycle modulation was inferred from the data-based part of the record, i.e. by employing the relationship between the solar cycle amplitude and the longer-term solar modulation levels (11-year averages) during last 500 years. This solar modulation record was scaled to the total solar irradiance (TSI) record by Schmidt et al. (2011,

2012) including longer-term trends in TSI (see "MEA (back)", Figure 8 in Schmidt et al. (2011)). This was done by linearly transforming the solar modulation to a TSI record in order to reproduce the long-term changes (11-year average) in TSI from the Maunder minimum to the most recent 50 years i.e. leading to a similar range of long-term TSI changes as suggested by Schmidt et al. (2011).

## 2.3    Climate reconstruction

We use winter seasonal means (Nov-Apr) for 8 Greenland ice cores (Vinther et al., 2010) for the period 1241-1970 CE (Table 2). All ice cores are synchronized via volcanic reference horizons, and the dating uncertainty is estimated to one year for the oldest parts of the records used here (Vinther et al., 2006). Under the assumption that $\delta^{18}$O in precipitation is a result of a number of processes mainly determined by atmospheric variability, we treat each year of the model run (see above) as a sample in a sampling space relating $\delta^{18}$O in precipitation with circulation, temperature, etc. By extracting precipitation weighted

winter seasonal means (Nov-Apr) of $\delta^{18}$O from the model at the 8 ice cores sites we can find the model years best matching the isotope pattern of each winter in the ice core data. In order not to over-fit noise in the ice core data (post depositional processes etc. (White et al., 1997)) we first perform a principal component analysis of the ice core $\delta^{18}$O and the model $\delta^{18}$O from grid cells covering the investigated ice core sites. We retain the first three PCs explaining a total of 60% and 97% of

variability in the ice core and model $\delta^{18}$O, respectively. The loadings of the 3 PCs for the model data match the loading of the ice core data well (Figure 1). Notice that this step is done without performing any selection of the model data, meaning that the modeled spatio-temporal variability of the $\delta^{18}$O in precipitation corresponds well to that of the ice core data. As we only explain part of the variability of the $\delta^{18}$O data using the first 3 PCs, we use an ensemble approach to take into account that the

matching of a given ice core $\delta^{18}$O pattern will result in a suite of well-matching model years. To match each year in the ice core data, the model data is evaluated using a $\chi^2$-measure between the 3 PCs of Greenland ice core $\delta^{18}$O and the 3 PCs of the modeled $\delta^{18}$O:

$$\chi^2_{Match} = \frac{1}{3} \sum_{k=1}^{3} (PC(k)_{model} - PC(k)_{icecore})^2 \qquad (1)$$

where $PC(k)_{model}$ and $PC(k)_{icecore}$ are the values from a given year of the normalized time series of model and ice core

$\delta^{18}$O PCs, respectively. Each model year is evaluated against each year of the ice core data and then sorted in ascending order of the quality of the fit. This creates 1201 (number of model years, see above) fits of model output for each year of the ice core data, i.e., 1201 resampled (with replacement) and sorted model years for the entire length of the ice core data (1241-1970 CE). We define the sorted model output as ensemble members, such that the best fitting model year, for each year of the ice core data, belongs to ensemble member 1, the second best fitting model years belong to ensemble member 2, and so forth.

Using a Chi-square goodness-of-fit test, with respect to the measure in Eq. 1, we evaluate the ensemble members against the PCs of Greenland ice core $\delta^{18}$O and reject model fits with likelihood p > 0.01 of not fitting the ice core data. This leaves us with 39 time series of reshuffled model data fitted to the ice core data. The temporal succession of the reshuffled model output does not resemble the order of the years in the original model run, and there are no systematic preferences of the method to pick certain time periods of the model run to match certain time periods of the reconstruction. For example if we exclude the

years 1851-2000 from the model run and perform the reconstruction using the remaining years, the reconstruction is almost identical to the reconstruction using all model years. This also means that the timing of the forcing used for the model run has no relation to the timing or impact of specific forcings in the reconstruction, i.e., the forcings of the model run effectively serve to produce enough range in order to match the simulation to the isotope variability of the ice cores. Since the timing of the forcings used for the model simulation is not a factor influencing the reconstruction, the model performance for the response

to forcings does not influence the reconstruction. We treat the 39 model fits as an ensemble solution fitting the model output to the ice cores $\delta^{18}$O, and we calculate the ensemble mean reconstructed $\delta^{18}$O and the standard deviation showing the ensemble spread. For the target climate variables (SLP, T2m) we extract the DJF ensemble mean corresponding to the reconstructed $\delta^{18}$O constituting the reconstruction of these variables. Note that using this approach the method is optimized to fit modeled $\delta^{18}$O to ice core $\delta^{18}$O records and, in contrast to presently existing reconstructions, it is not calibrated to match observations of any of

the target variables of the reconstruction (i.e. SLP or T2m).

As a first test we correlate the reconstructed $\delta^{18}$O to winter means of ice core data from 20 cores covering the last 200 years of the reconstruction, i.e. including ice core data not part of the reconstruction. The correlation shows a spread between 0.4 and 0.9 with the highest correlations found for the high accumulation sites, and where multiple ice cores for the same site are used for the reconstruction. This supports the idea that the skill of the model fit to the seasonal ice core data is largely limited by the

signal-to-noise ratio of the ice core data (Figure S1). High accumulation sites are generally less sensitive to wind scoring and post depositional diffusion. Despite using the variability of the PCs to fit the model output to the ice core data, the reconstructed range of $\delta^{18}O$ matches the ice core data well (Figure S2). Furthermore we test the fit of the reshuffled model $\delta^{18}O$ to the ice core data of the 8 sites to investigate if any time periods stand out, as well as the mean fit of the method across the whole period

of the reconstruction. We find no trends in the performance in terms of fitting the modeled PCs to the ice core data PCs. As the method performs equally well during any period as during the instrumental period (post 1850) in terms of matching the ice core isotope variability, it is likely that our reconstruction of SLP and T2m is equally valid for any period as it is for the instrumental period. This conclusion can be drawn as the reconstruction is not calibrated to either SLP or T2m, and is only constrained by the ice core isotope variability.

**2.4   Statistical tests and filtering of data**

We test the significance for anomalies of climate field variables with a two-tailed Student's t-test. For low-pass and band-pass filtering of data series we use a Fast Fourier Transform approach if data is used for correlation analysis. In Figure 2 b we use a 'loess' filter to smooth the data for visualization of the multi-decadal variability. When calculating significance for correlations of filtered data we use the method by Ebisuzaki (1997) to take auto-correlation into account.

**3   Results**

**3.1   Evaluation of climate reconstruction**

Comparing to the 20th Century Reanalysis (Compo et al., 2011) (20CR) our reconstruction shows skill for SLP and T2m in the North Atlantic region (Figure 2 a, b), the main mode of atmospheric circulation (NAO) as well as secondary circulation modes (Figure 2 c, d and Table 3). This is a completely independent test of the reconstruction as the reconstruction has not

be calibrated to reanalysis data or observations. For T2m the pattern of significant correlations with the 20CR data can be associated with the main circulation modes (Figure S3 and S4), albeit with decreasing skill with the distance to the ice cores. We interpret the high skill near Greenland as being due to the direct physical connection between the local temperature in Greenland and the temperature along the path of the vapor, and the isotopic signal in Greenland ice cores. Contrary to previous millennial scale reconstructions, our reconstructed NAO shows similar strength of year-to-year variability as the observed

NAO indicating that the reconstruction preserves the known characteristic variability of the NAO (Figure 2 d, e, Figure 3 a). In addition to capturing the NAO, our reconstruction has skill in representing Atlantic ridge/trough-type variability as projected on PC2 of SLP over the North Atlantic. The correlation of the reconstructed SLP PC2 and PC2 of the 20CR SLP is 0.24 (p < 0.01) and increases to 0.53 (p < 0.01) on decadal time scales (Table 3). However, comparing the SLP patterns of PC2 in our reconstruction and reanalysis (Figure S3) implies that the variability captured by in PC2 of the reconstruction is likely split

between PC2 and PC3 in the reanalysis. Indeed, the reconstructed PC2 of SLP is also correlated to PC3 of the reanalysis SLP (corr. = 0.19, p < 0.01, Table 3). Correlating the reconstructed PC2 of SLP against the sum of the reanalysis PC2 and PC3

shows increased correlations between the reconstruction and reanalysis, in particularly on decadal and bi-decadal time scales (Table 3). This indicates that the variability projected on PC2 and PC3 of the reanalysis data is partly summarized in PC2 of the reconstruction. The difference in distribution of the variability on the PCs in the reconstruction and reanalysis is likely due to i) the intrinsic variability of the model use for the reconstruction and ii) the reconstruction only captures the North Atlantic variability as recorded in the isotopic composition of the Greenland ice core data.

The reconstructed NAO shows strong multi-decadal variability, while no major trends are found on centennial time scales, as opposed to the reconstruction by Trouet et al. (2009), and in agreement with the NAO reconstructions by Ortega et al. (2015) (Figure 3). The NAO reconstructions by Ortega et al. (2015) consists of two multi-proxy NAO reconstructions. Of these reconstructions one is calibration-constrained (NAOcc) and the other model-constrained (NAOmc) (Figure 3 b), where NAOmc only uses proxies from sites that were estimated from model simulations to have a stable relation to the NAO. It should be noted that both the reconstruction by Ortega et al. (2015) and this study use some of the same Greenland ice core data (Crete, DYE-3, GRIP), which obviously could lead to a correspondence in variability. We find the NAO in our SLP reconstruction has best correspondence with NAOmc for interannual variability, while for multi-decadal time scales prior to the instrumental period, there is little coherency between our reconstruction and both NAOmc and NAOcc (Table 3). This lack of coherence of multi-decadal variability is similar to the aforementioned divergence between previous NAO reconstructions. For an independent comparison we used reconstructed NAO and gridded reconstructed SLP over Europe (Luterbacher et al., 2001), as well as gridded reconstructed temperature (Luterbacher et al., 2004) over Europe. However, we restricted the comparison to 1659-1970 due to the methodological differences in the reconstructions for Europe prior and post 1659, and the use of Greenland ice core data for the European temperature reconstruction covering 1500-1658. For the NAO reconstruction by Luterbacher et al. (2001) our reconstruction shows slightly lower correlation on interannnual time-scales compared to the correlation to NAOmc, but similar correlation on decadal to multi-decadal time-scales (Table 3). The comparison between our reconstruction and these reconstructions for Europe shows similar pattern and correlation levels for SLP as with the 20CR data, and moderate, but significant, correlations for temperature (Figure S5).

Our reconstructed NAO shows higher correlation (corr. = 0.52, $p < 0.01$) to the observed NAO (Jones et al., 1997) (DJF, 1824-1970) than NAOmc and NAOcc, which have correlations to the observed NAO of 0.46 and 0.47, respectively. Even more important, the skill of our reconstructed NAO is achieved without calibrating to the observed NAO. In summary, we think that our reconstruction is the most suitable for analyzing the influence of volcanic eruptions and solar activity on circulation, because i) our reconstruction not only has good skill for the NAO, but also for the secondary modes of circulation, which, as we will show later, is crucial for investigating the impact of solar forcing ii) high frequency variability is preserved, making it possible better to detect rapid shifts in circulation after volcanic eruptions, and finally iii) our reconstruction is not calibrated to observed SLP or T2m, making it free of biases that could arise from tuning a reconstruction to observations during the instrumental era.

## 3.2   Response of atmospheric circulation to external forcing

In this section we investigate the reconstructed response in SLP and temperature to major tropical volcanic eruptions and solar variability. The mean post eruption SLP and T2m anomalies in response to 12 major tropical eruptions show the characteristics of a positive NAO (Figure 4 a, b). On average, we find a significant positive NAO response during the five consecutive winters

following the eruptions (Figure 4 c). Performing the same analysis on NAOmc yields a similar response, while not reaching as high significance levels and persistence as our reconstruction, likely due to the attenuated year-to-year variability of NAOmc (Figure 2 c & 4 c, Figure S6). Due to the short time span between some of the volcanic eruptions it is not possible to consistently analyze any longer term response than five years for single eruptions, since it would limit the number of eruptions and, hence, the robustness of the statistical analysis. We analyzed the decadal to multi-decadal response to volcanic forcing by

calculating the correlation between reconstructed NAO and volcanic forcing from tropical eruptions (Toohey and Sigl, 2017) after filtering both data series with a band-pass filter. The correlation analysis of filtered data estimates the combined effect of the eruptions during the reconstructed time frame. Due to the very abrupt nature of the volcanic forcing, heavy filtering can introduce artificial forcing prior to the onset of the actual forcing. We find that smoothing the volcanic forcing data using a low-pass filter with no less than 1/10 cycle per year only has negligible effects on the analysis. Performing a time-lag correla-

tion analysis on the band-pass filtered data (1/10 to 1/100 cycles per year) we obtain significant correlations (p < 0.01) with the reconstructed NAO lagging the volcanic forcing 1 to 6 years (Figure S7). This corresponds well to the results of the analysis of the mean NAO response for the 12 major tropical eruptions. Maximum correlation is reached at time lags of 3 to 4 years with a correlation of 0.33. This shows the cumulative effect of several volcanic eruptions which can cause trends in the NAO on longer timescales than single eruptions.

For the analysis of solar influences on circulation we analyzed the average response to the 11-year solar cycle and the multi-decadal to centennial solar variability. We calculated the difference between reconstructed SLP and T2m for years of low and high solar activity using the annual sunspot number (Clette and Lefèvre, 2016) (1700-1970) and a [14]C-based solar reconstruction (1241-1970) (see Sect. 2.2) for the short term and long term cycles, respectively. The response to the 11-year solar cycle (solar low minus high) is a high pressure over Scandinavia corresponding well to the pattern found for reanalysis data (Figure 5

a, d and Figure 6 a, c). This anomalous high pressure could be due to increased frequency in Scandinavian blockings, which has been shown to impact Greenland $\delta^{18}$O (Ortega et al., 2014; Rimbu et al., 2017). Investigating the time lagged response to the 11-year solar cycle we find the strongest response in reconstructed SLP and T2m when lagging the solar forcing with 5 years, which also matches the time lagged pattern found in reanalysis data (Figure 5 b, e and Figure 6 b, d). This pattern projects on PC2 of the reconstructed SLP, also with the strongest correlation between forcing and response when lagging the sunspot data

with 5 years. Taking this time lag into account yields a consistent relationship between solar forcing and PC2 of reconstructed SLP on decadal to multi-decaldal time scales (Figure 7, Table 4). The 5-year lagged response in circulation is not simply due to the response to the solar maximum approximately half a cycle later in the 11-year solar cycle, but a reinforced response. This is most clearly seen in the stronger correlations for the time lagged response, both for the original data and the filtered data (Table 4, Figure 7 and Figure S8). The relation between PC2 of reconstructed SLP and solar forcing persists also for centennial

variability, which is seen by comparing the circulation response to the [14]C-based solar reconstruction (Table 4, Figure S9). The pattern of the response in SLP to the long-term solar minima is an Atlantic ridge-type pattern (anomalous high south of Greenland), which also projects on PC2 of SLP, with an associated cooling pattern for the western North Atlantic (Figure 5 c, f). Compared to the 5-year lagged response to the 11-year cycle this pattern has the strongest response in SLP south of Greenland, with a similar, but more widespread cooling in the eastern North Atlantic. Even though the SLP response looks slightly different for short and long-term solar forcing variations, the main feature: a wave structure over the North Atlantic and Scandinavia, is consistent (Figure 5 b, c). This similarity in the 5-year lagged and long-term response can also be seen in the patterns of the temperature anomalies (Figure 5 e, f). The temperature response to the long-term solar minima is a cooling across Greenland, Iceland and western Europe during solar minima (Figure 5 f). This cooling pattern corresponds well to the suggested cooling during the Little Ice Age in proxy records from Greenland (Stuiver et al., 1997), Iceland (Moffa-Sanchez et al., 2014) and Europe (Luterbacher et al., 2004). A NAO-type response to long-term solar forcing would give opposing temperature responses in Greenland and Europe, which is not the case. We find no consistent relation between our reconstructed NAO and solar forcing. Instead we would like to stress the importance of the connection between solar activity and the secondary circulation patterns, which possibly shows the main response to solar forcing on decadal to centennial time scales, with correlations of 0.29 (p < 0.01) and 0.6 (p < 0.01), respectively.

## 4    Discussion and conclusions

The model simulation used for our reconstructed translates the climate variability recorded in the Greenland ice cores to climate variability in the North Atlantic region. In the initial test of the isotope variability it is shown, that the spatio-temporal $\delta^{18}O$ variability of the ice cores is well represented by the model (Figure 1). This is a fundamental prerequisite which allows us to match the modeled $\delta^{18}O$ to the ice core $\delta^{18}O$ year-by-year. While the skill of the reconstruction is higher in the vicinity around Greenland, the reconstruction shows significant correlations to reanalysis data wide spread across the North Atlantic region. This skill depends on i) the integrative nature of the $\delta^{18}O$ as recorded in the ice cores, and represented by the modeled $\delta^{18}O$ ii) the modeled atmospheric teleconnection patterns in terms of temperature and circulation, and iii) how these patterns are connected to modeled $\delta^{18}O$ for Greenland. Clearly, the reconstruction is strongly dependent on the climate model when it comes to whether or not it is possible at all to use our method, and when it comes to the skill of reconstructed spatial patterns. The resolution of our model model simulation is relatively course and using a higher resolution simulation could improve the representation of several processes. For example, vapor transport to dry polar regions is often inhibited in models with courser resolution, resulting in too little precipitation in the interior of ice sheets and a positive bias in $\delta^{18}O$ (Masson-Delmotte et al., 2008; Sjolte et al., 2011). This is related to cloud parameterizations and course resolution models having difficulties in explicitly representing frontal zones in connection with synoptic weather systems. The orography in course resolution models is also more smooth, loosing orographical features such as the southern dome of the Greenland ice sheet, which also affects atmospheric circulation and small scale spatial variability. In our approach we match the modeled PCs of $\delta^{18}O$, meaning that we are matching regional scale patterns in $\delta^{18}O$, which partly addresses the problem of matching course model output to site

specific proxy data. However, having a higher resolution model simulation could for example improve the spatio-temporal representation of Greenland $\delta^{18}$O, allowing more than 3 PCs to be fitted, and generally giving a better representation of temperature, pressure and precipitation in the reconstruction. For reasons discussed above, it would be desirable using different GCMs to test for model dependencies of the reconstruction, as well as testing for added value of ensemble reconstructions

with several different GCMs. Doing these tests is presently limited by the availability of millennium length simulations using isotope enabled GCMs.

We selected the proxy records for this study based on the criterion of having seasonal resolution, small dating uncertainty, a long time span and a wide regional spread. In order to provide a quantitative link to the isotope enabled GCM we selected only isotope-based proxies. For the time being, this leaves us with the 8 Greenland ice cores used in this study. Other seasonal

resolution ice cores from Greenland are available, but only covering a limited time span, and comparing to these cores shows that the reconstructed $\delta^{18}$O also compares well to the isotopic variability at these sites (Figure S2). However, including more Greenland ice cores of similar quality would generally improve the signal to noise ratio of the reconstruction, and such records should be included if available for subsequent studies. Obtaining seasonal resolution in ice core data is mainly limited by the accumulation rate and seasonality of precipitation, which depends on the regional climate setting of the drill site (Vinther et al.,

2010; Zheng et al., 2018). Including other archives than ice cores would give a more widespread regional coverage, potentially providing better constraints on circulation patterns and climate trends. Some oxygen isotope records from tree rings in Sweden (e.g., Edwards et al., 2017) and speleothems from the European alps (e.g., de Jong et al., 2013)) covering the past millennium primarily reflect winter climate conditions. Both records in these examples have 5-year resolution, and the speleothem record has hiatuses, which reflect some of the challenges in using these proxy records. However, there could be benefits of using a

larger selection of data, despite the different temporal resolution (Steiger and Hakim, 2016).

The comparison between the response to volcanic forcing between our reconstruction and NAOmc (Figure 3 c) shows that the mean response look qualitatively very similar in the two reconstructions. However, as already mentioned, due to the preserved high frequency variability our reconstruction shows both a more immediate and persistent NAO response to volcanoes. This underlines the importance of producing climate reconstructions that do preserve high frequency variability, in particular

if the reconstruction is used as baseline for model evaluation. In our analysis of the reconstructed response to volcanic eruptions we choose eruptions larger than or of similar magnitude as the 1991 Pinatubo eruption ($< $ -6 Wm$^{-2}$). As discussed by Swingedouw et al. (2017) climate effects of smaller eruptions can be difficult to detect due to stochastic climate variability. We find that we can detect the an impact on reconstructed NAO from tropical eruptions selected in the range from -4 Wm$^{-2}$ to -8 Wm$^{-2}$ (Sigl et al., 2015), yielding a significant positive NAO one year after the eruptions, on average. This appears to be the

limit of detection for our reconstruction, possibly owing both to the partly stochastic variability of the NAO and to noise in the reconstruction.

Model studies of the volcanic response to major tropical eruptions during the past millennium show a large spread in the modeled NAO/AO response, with either no consistent response or 1-2 years of significant response (Swingedouw et al., 2017; Zambri et al., 2017). In contrast to this we find a clear tendency for positive NAO for the five consecutive winters following the

year of eruption as an average response to the 12 largest tropical eruptions during 1241-1970 CE. An immediate strengthening

of the polar vortex following eruptions is in agreement with the observed response of atmospheric circulation (Driscoll et al., 2012), which then translates to a positive NAO as the stratospheric anomaly propagates to the troposphere. The presence of volcanic aerosols gradually tails off of during the first 2-3 years (Crowley and Unterman, 2013) and a more sustained positive NAO for up to five years could be explained via a positive ocean feedback through a tri-pole SST response to the strongly anomalous positive NAO (Cayan, 1992). Ongoing efforts to improve the simulation of volcanic forcing and response could help close the gap between models and observations as well as reconstructions (Zanchettin et al., 2016).

It has been suggested that the observed increase in blocking frequency over the North Atlantic in response to solar minima (Woollings et al., 2010) is coupled to a weakening of the polar night jet in response to a weaker stratospheric equator-pole temperature gradient. This mechanism could be in play on both decadal and centennial time scales. A recent study investigated the response in circulation to solar activity using a regression-based analysis between sunspot data and gridded observed SLP and SST data (Gray et al., 2016). The authors analyzed the time lagged response to solar forcing, and found that the solar response could be explained via two mechanisms (Gray et al., 2016). One involving the aforementioned stratosphere-troposphere coupling acting on time lags of 0-2 years, and one for time lags of 3-4 years involving ocean temperature anomalies being stored beneath the mixed layer and reinforced from the previous winter (Gray et al., 2016). The reinforcement of SST anomalies from year-to-year has also been shown in a simulation of the response to the solar 11-year cycle (Andrews et al., 2015). Such a mechanism could be the cause of the time lag we see in the reconstructed response to solar forcing, although we get the maximum response at 5-year time lag, compared to the 3-4 year time lag found in observations (Gray et al., 2016). However, this difference could be due to differences in the methodologies of the analysis of the response to solar forcing, and that the aforementioned study (Gray et al., 2016) is focusing on the NAO-like response seen in their analysis. A possible mechanism based on our findings is as follows. An initial increase in atmospheric blockings weakens the subpolar gyre (Moffa-Sanchez et al., 2014) (SPG), thereby decreasing the heat transport to the north-western North Atlantic giving favorable conditions for mid-Atlantic blocking. This pattern is reinforced year-by-year and the main atmospheric response shifts to the node of PC2 of SLP south of Greenland under sustained forcing conditions on longer time scales (Figure 5 c). A recent model study (Moreno-Chamarro et al., 2017) suggested that the cooling during the Little Ice Age was connected to a weakening in the SPG, sustained by way of atmosphere-ocean feedbacks. Although the authors do not related this to solar forcing, but with preconditioned initial model variability, the anomalies in SLP and temperature associated with the weakening of the SPG are very similar to the reconstructed pattern of the response to long-term solar forcing. One explanation could be that low solar activity is the preconditioning factor in reality, causing the response to solar forcing seen in our reconstruction, while the climate response to solar forcing might not be fully captured by the MPI-ESM (Mitchell et al., 2015) used by Moreno-Chamarro et al. (2017). Our study of the reconstructed North Atlantic winter circulation shows a complex response to solar forcing which is, in contrast to a prevalent hypothesis, not directly linked to the NAO. The complexity is also reflected in a non-uniform temperature response to solar forcing, with both regional warming and cooling. This also means that part of this signal will be smoothed out if such analysis is carried out on hemispherical mean temperature (e.g., Schurer et al., 2015). In our study we do not exclude that there could be an influence of the solar 11-year cycle on NAO. However, unlike for PC2 of reconstructed SLP, we find no consistent relationship between reconstructed NAO and solar forcing across multiple time scales. Furthermore, the

results suggest that sustained longer-term solar forcing leads to a shift in the atmospheric circulation response compared to the response to the short-term forcing, possibly due to feedback processes involving the ocean integrating the long term effects of anomalous atmospheric circulation.

Our study presents a new climate reconstruction of SLP and temperature for the North Atlantic region. The reconstruction not only resolves the first mode of atmospheric circulation (PC1), the NAO, but also captures the second mode (PC2), referred to as the Eastern Atlantic pattern. In the analysis of our reconstruction we find that solar and volcanic forcing impacts different modes of the atmospheric circulation during winter, which can aid to separate the regional effects of forcings and understand the underlying mechanisms. The reconstructed response to forcings can also serve as a baseline for climate model evaluation. Although atmospheric variability to a large extent is a stochastic process, the variability in our reconstruction also shows an over all significant impact of forcings. The squared correlation coefficient can provide an estimate for explained variance of the external forcings. Using this approach, tropical volcanic forcing accounts for about 10% of the decadal to multi-decadal variability of the reconstructed NAO, while solar forcing accounts for about 40% of the variability of PC2 of reconstructed SLP on centennial timescales.

*Data availability.* The time series of PC1 and PC2 of reconstructed SLP in the North Atlantic Region have been submitted to the PANGAEA open access data library. To obtain the complete climate field reconstructions of SLP and T2m, or the ECHAM5/MPI-OM simulation, please contact the corresponding author.

*Author contributions.* J.S. developed the method, performed the analysis, conducted the model simulation and wrote the first version of the manuscript. C.S. initiated the study and contributed to setting up the model simulation. F.A. contributed to the method development and analysis. B.V. provided seasonal ice core data. M.W. provided technical support for model simulation and access to climate model. G.L. provided insights on model setup. R.M. contributed with solar activity reconstruction and provided insight on solar forcing of climate. All authors discussed and edited the manuscript.

*Competing interests.* The authors declare no competing interests.

*Acknowledgements.* This work was supported by the Swedish Research Council (grant DNR2011-5418 & DNR2013-8421 to RM), the Crafoord foundation and the strategic research program of ModEling the Regional and Global Earth system (MERGE) hosted by the Faculty of Science at Lund University. The model simulation was carried out at the AWI Computer and Data Center, Bremerhaven.

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

**Table 1.** List of forcings for the ECHAM5/MPI-OM model simulation.

| | |
|---|---|
| Greenhouse gases ($CO_2$, $CH_4$, $N_2O$) | MacFarling Meure et al. (2006) |
| Greenhouse gases (historical, anthropogenic) | Marland et al. (2003) |
| Ozone | Climatology of Paul et al. (1998) |
| Volcanic aerosols | Crowley et al. (2008) |
| Aerosol forcing | Background from Tanre et al. (1984) and post 1850 variations by Lefohn et al. (1999) |
| Total solar irradiance | Based on Muscheler et al. (2016, 2007) (see also Methods) |
| Land-use | Pongratz et al. (2008) with vegetation from Jungclaus et al. (2010) E1 ensemble member mil0010 |
| Orbital forcing | Variation Seculaires des Orbites Planetaires (VSOP) analytical solution by Bretagnon and Francou (1988) |

**Table 2.** Site details of ice core records used for the reconstruction (Vinther et al., 2010).(*) BC indicates the at the core was drilled to bedrock.

| Drill site | Lat. ($^o$N) | Long. ($^o$W) | Elevation (m a.s.l.) | Acc. Rate (m ice/yr) | Time span |
|---|---|---|---|---|---|
| Crete | 71.12 | 37.32 | 3172 | 0.289 | 551-1974 |
| DYE-3 71 | 65.18 | 43.83 | 2480 | 0.56 | 1239-1971 |
| DYE-3 79 | 65.18 | 43.83 | 2480 | 0.56 | BC*-1979 |
| GRIP 89-1 | 72.58 | 37.64 | 3238 | 0.23 | 918-1989 |
| GRIP 89-3 | 72.58 | 37.64 | 3238 | 0.23 | BC*-1989 |
| GRIP 93 | 72.58 | 37.64 | 3238 | 0.23 | 1062-1993 |
| Milcent | 70.30 | 44.50 | 2410 | 0.54 | 1173-1973 |
| Renland | 71.27 | 26.73 | 2350 | 0.50 | BC*-1988 |

**Table 3.** Correlations of reconstructed NAO and PC2 of reconstructed SLP, and observed NAO, 20CR PC2 and PC3 of SLP, as well as NAO reconstructions by Ortega et al. (2015) and Luterbacher et al. (2004). All correlations are for detrended data, and p-values are calculated with the random-phase test by Ebisuzaki (1997) to take into account auto-correlation.

| | Annual (DJF) | 10-year low-pass | 20-year low-pass |
|---|---|---|---|
| Corr($NAO_{recon}$, HadCRU) 1824-1970 | 0.52 (p<0.01) | 0.68 (p<0.01) | 0.70 (p<0.01) |
| Corr($NAO_{recon}$, $NAO_{20CR}$) 1851-1970 | 0.44 (p<0.01) | 0.44 (p<0.01) | 0.46 (p<0.01) |
| Corr($SLP_{PC2-recon}$, $SLP_{PC2-20CR}$) 1851-1970 | 0.24 (p<0.01) | 0.53 (p<0.01) | 0.58 (p<0.01) |
| Corr($SLP_{PC2-recon}$, $SLP_{PC3-20CR}$) 1851-1970 | 0.19 (p<0.01) | 0.57 (p<0.01) | 0.66 (p<0.01) |
| Corr($SLP_{PC2-recon}$, $SLP_{PC2+PC3-20CR}$) 1851-1970 | 0.30 (p<0.01) | 0.67 (p<0.01) | 0.84 (p<0.01) |
| Corr($NAO_{recon}$, Ortega et al. MC) 1241-1969 | 0.49 (p<0.01) | 0.43 (p<0.01) | 0.37 (p<0.05) |
| Corr($NAO_{recon}$, Ortega et al. MC) 1241-1820 | 0.47 (p<0.01) | 0.36 (p<0.05) | 0.15 (p<0.2) |
| Corr($NAO_{recon}$, Ortega et al. CC) 1241-1969 | 0.36 (p<0.01) | 0.35 (p<0.01) | 0.40 (p<0.01) |
| Corr($NAO_{recon}$, Ortega et al. CC) 1241-1820 | 0.29 (p<0.01) | 0.21 (p<0.05) | 0.12 (p<0.2) |
| Corr($NAO_{recon}$, Luterbacher) 1659-1970 | 0.34 (p<0.01) | 0.39 (p<0.01) | 0.40 (p<0.01) |

**Table 4.** Correlation between solar forcing and PC2 of SLP, with and without time lag. The first column indicate which data is used, and if any filtering is done to the data. Second column is correlation coefficients between solar forcing and PC2 of reconstructed SLP, with solar forcing either being represented by sunspot number or $^{14}$C data. The third column is correlation coefficients between solar forcing and PC2 of reconstructed SLP, with solar forcing represented by sunspot number shifted for a lag of 5 years. All correlations are for detrended data, and p-values are calculated with the random-phase test by Ebisuzaki (1997) to take into account auto-correlation.

| | No time lag | 5-year time lag |
|---|---|---|
| Corr(Recon. PC2 SLP, sunspots) 1700-1970 | -0.06 (p>0.1) | 0.20 (p<0.01) |
| 5-year low-pass filtered data: Corr(Recon. PC2 SLP, sunspots) 1700-1970 | -0.07 (p>0.1) | 0.29 (p<0.01) |
| 20-year low-pass filtered data: Corr(Recon. PC2 SLP, sunspots) 1700-1970 | 0.30 (p<0.05) | 0.53 (p<0.01) |
| 20-500 year band-pass filtered data: Corr(Recon. PC2 SLP, solar activity ($^{14}$C)) 1241-1970 | 0.30 (p<0.01) | - |
| 60-500 year band-pass filtered data: Corr(Recon. PC2 SLP, solar activity ($^{14}$C)) 1241-1970 | 0.60 (p<0.01) | - |

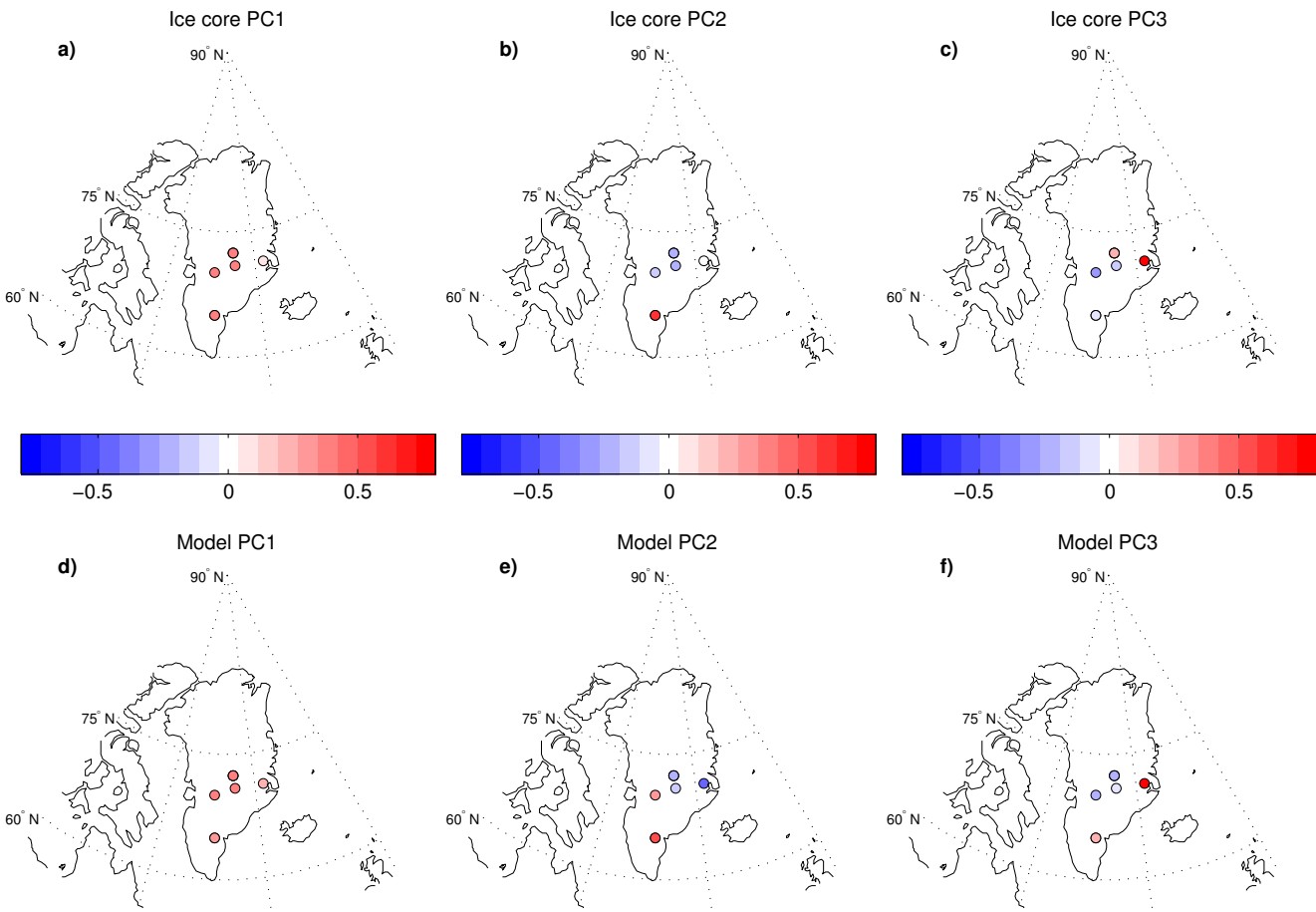

**Figure 1.** Spatial patterns of the three main modes of variability in the $\delta^{18}$O ice core records used in this study and the modeled modes of variability at these sites. The pattern of the loadings on ice core $\delta^{18}$O PCs are shown in (**a, b, c**) and modeled loadings on $\delta^{18}$O PCs at the ice core sites are shown in (**d, e, f**).

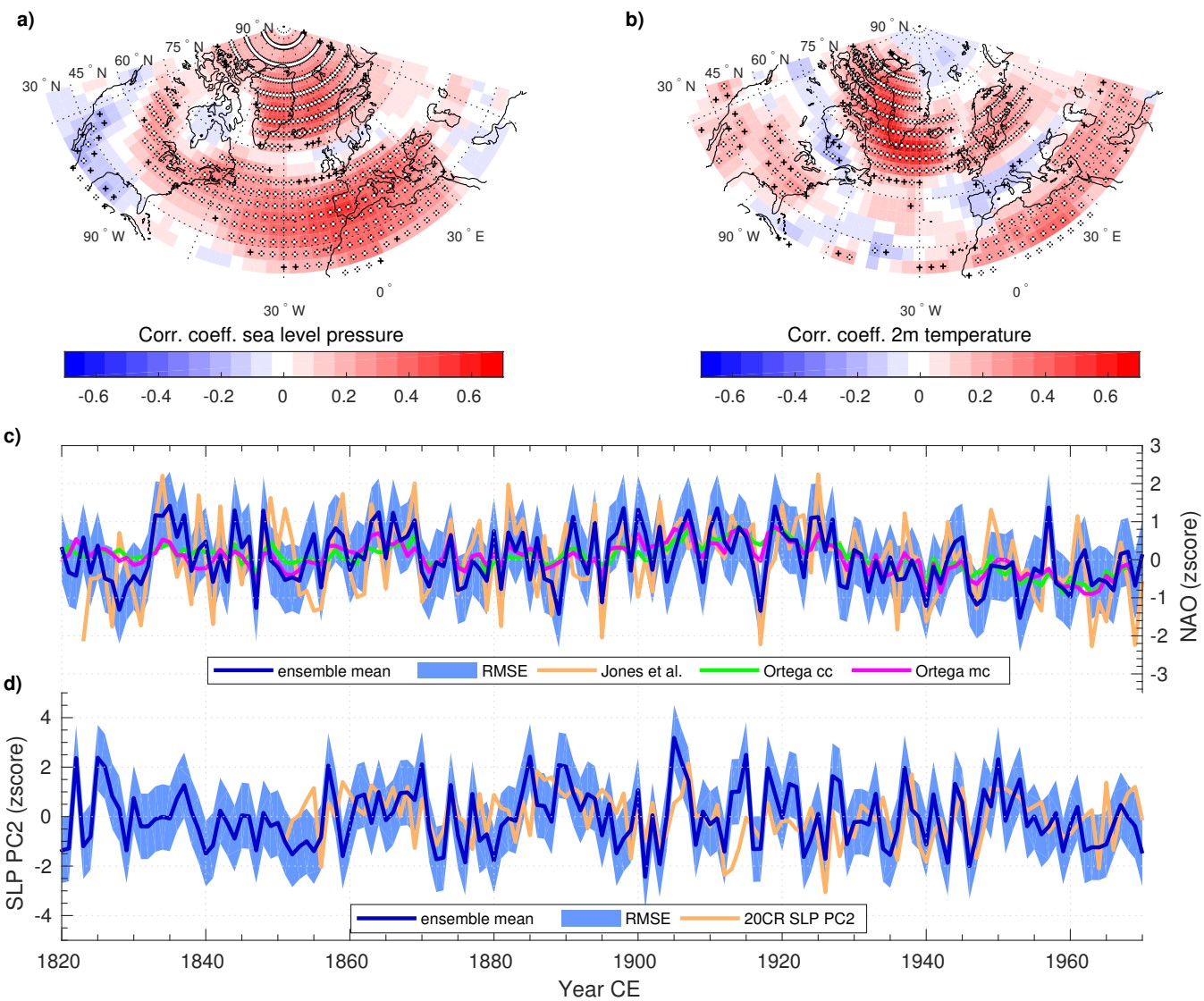

**Figure 2.** Evaluation of the winter circulation reconstruction. Grid point correlation between reconstructed DJF SLP (**a**) and T2m (**b**), and reanalysis data (Compo et al., 2011) (1851-1970) interpolated to the model grid (lat. x lon. ∼3.75° x 3.75°). The white stippling indicates significance p < 0.05, and black stippling indicates significance p < 0.1. **c** Ensemble mean reconstructed DJF NAO (PC1 of reconstructed DJF SLP (Hurrell et al., 2003)) with Root Mean Square Error (RMSE) compared to observed DJF NAO (Jones et al., 1997), NAOcc and NAOmc by Ortega et al. (2015). (**d**) Ensemble mean PC2 of reconstructed DJF SLP with RMSE compared to PC2 of reanalysis DJF SLP (Compo et al., 2011).

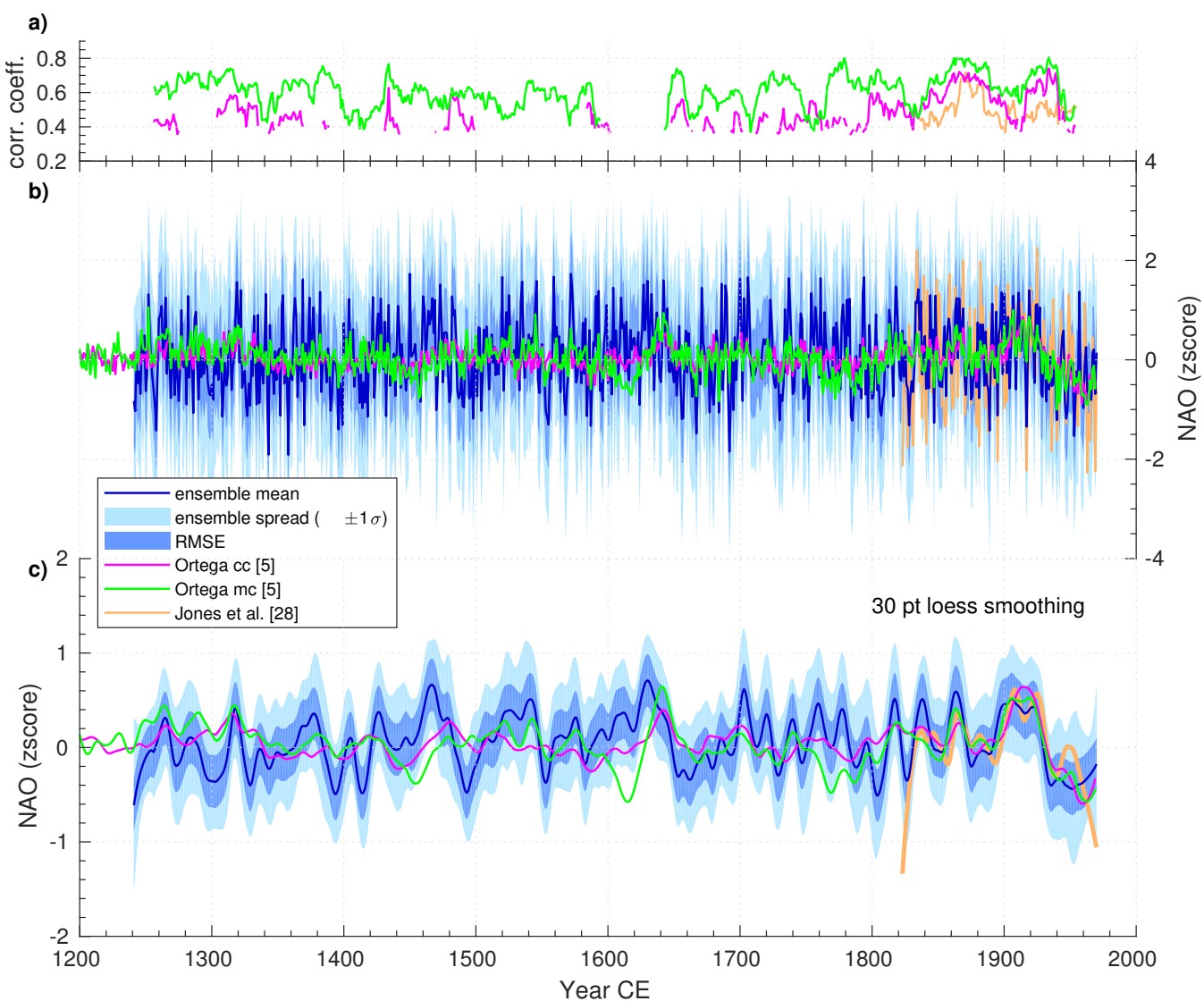

**Figure 3.** Comparison of instrumental NAO and proxy-based NAO reconstructions. **a** Moving 31-point correlation between reconstructed NAO from this study and NAOcc (magenta), NAOmc (green) (Ortega et al., 2015) and observed NAO (yellow) (Jones et al., 1997). Only significant correlations are plotted (p < 0.05). **a** Ensemble mean reconstructed NAO (PC1 of reconstructed SLP (Hurrell et al., 2003)) with error estimated by ensemble spread and RMSE, compared to observed NAO (Jones et al., 1997) and NAO reconstructions by Ortega et al. (2015). The amplitude of all time series are scaled to fit the decadal variability of the observed NAO. **c** Same as **b**, except filtered with a 30 point 'loess' filter.

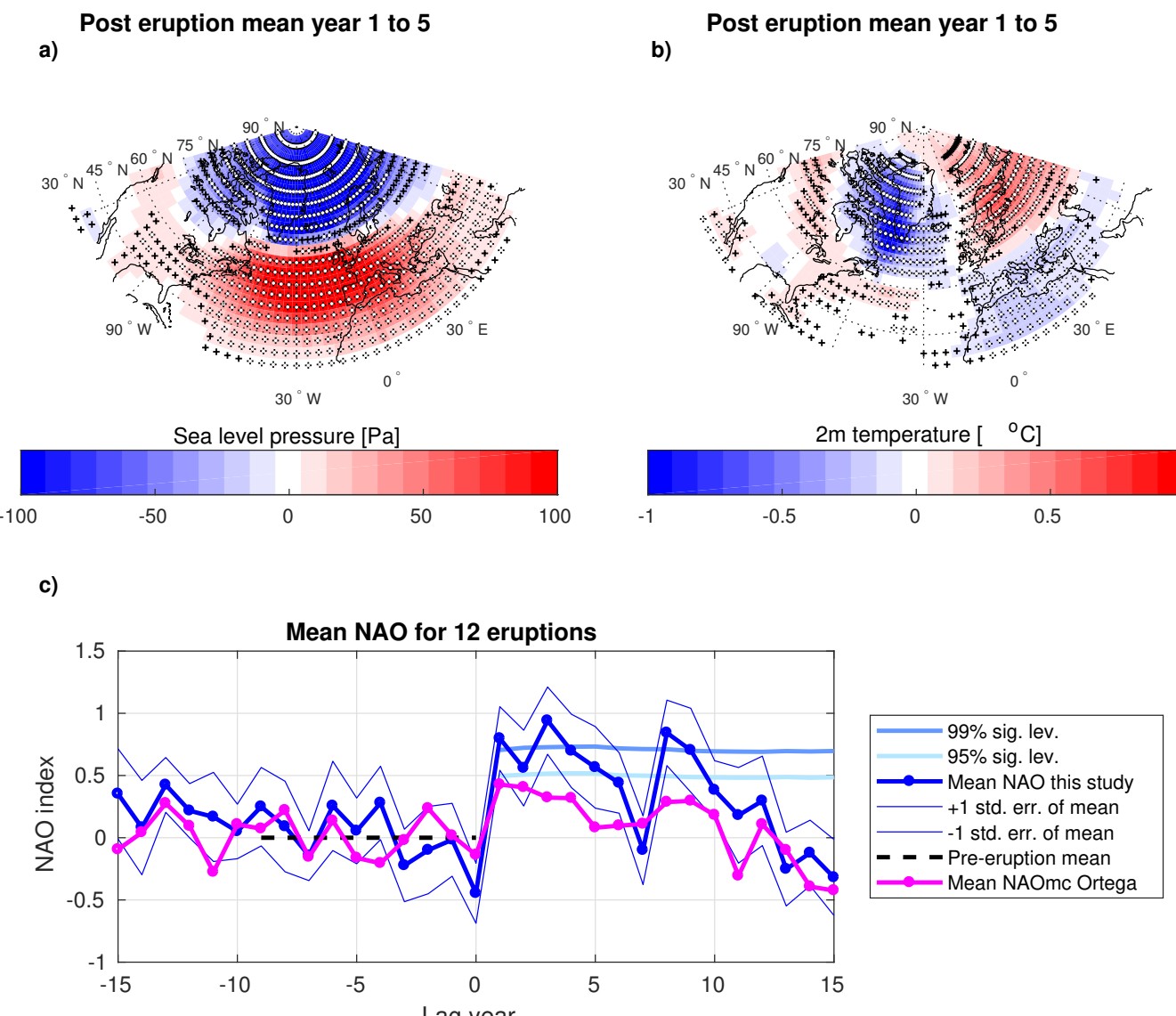

**Figure 4.** Superimposed epoch analysis of the mean response in atmospheric circulation to the 12 largest tropical volcanic eruptions (Sigl et al., 2015) (Table S1). The response in SLP and T2m is normalized to the mean fields of the 10 years preceding the eruption. **a** Mean DJF SLP anomalies [Pa] for the first five post eruption years. **b** Mean DJF T2m anomalies [$^{o}$C] for the first five post eruption years. The white stippling indicates significant anomalies p < 0.01, and black stippling indicates significant anomalies p < 0.05 (two-tailed Student's *t*-test). **c** Mean response in reconstructed NAO (blue) with the time series normalized to the mean NAO of the 10 years preceding the eruption. For comparison the same analysis is carried out for the NAOmc reconstruction (magenta) by Ortega et al. (2015). The significance levels in **c** are estimated from 100,000 random samples of 12 years drawn from the reconstructed NAO. See Figure S6 for significance levels for NAOmc.

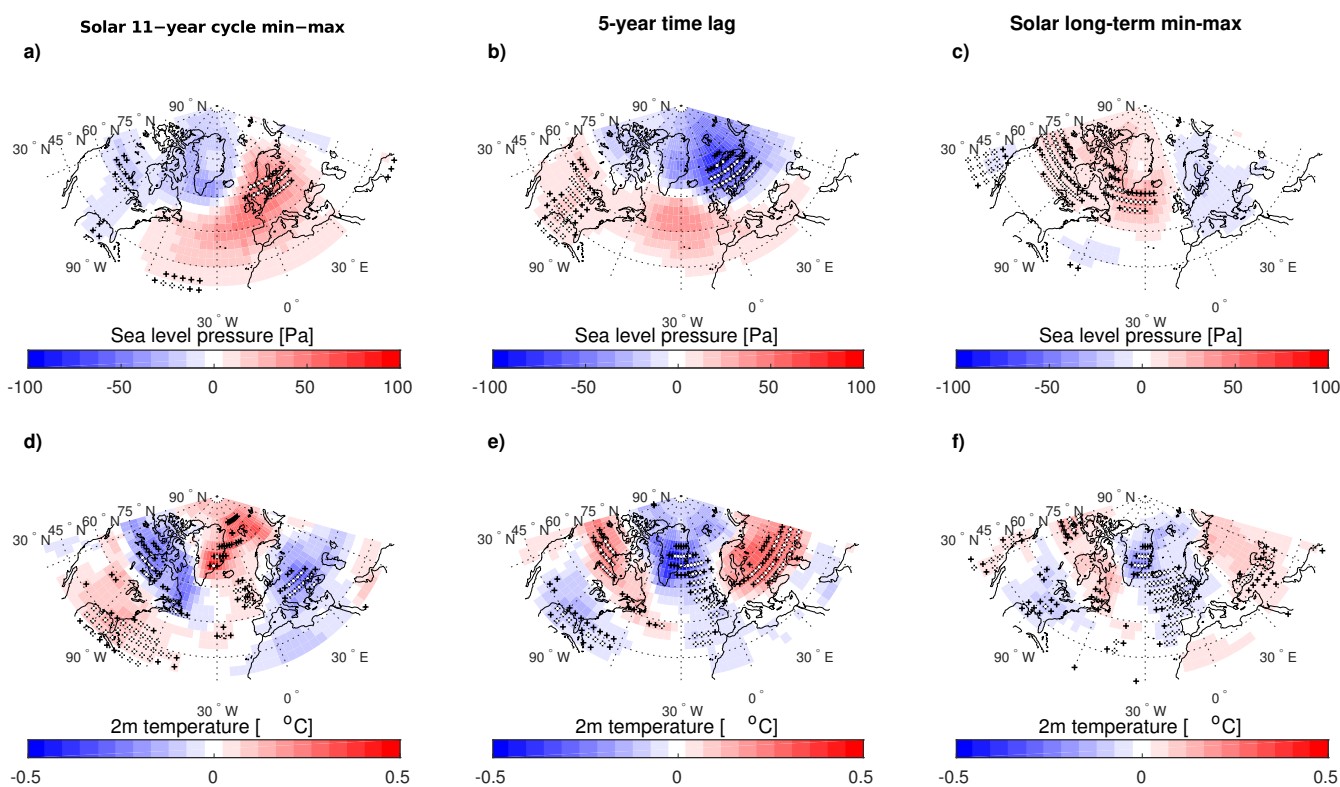

**Figure 5.** Reconstructed atmospheric response to solar forcing. **a** DJF SLP anomalies [Pa] in response to the 11-year solar cycle (solar min. minus solar max. defined in Figure S10). **b** DJF SLP anomalies [Pa] in response to the 5-year lagged 11-year solar cycle (solar min. minus solar max.). **c** DJF SLP anomalies [Pa] in response to the long-term solar forcing (solar min. minus solar max. defined in Figure S11). **d, e, f** corresponding figures to **a, b, c**, but for T2m [$^oC$]. The white stippling indicates significant anomalies $p < 0.05$, and black stippling indicates significant anomalies $p < 0.1$ (two-tailed Student's *t*-test).

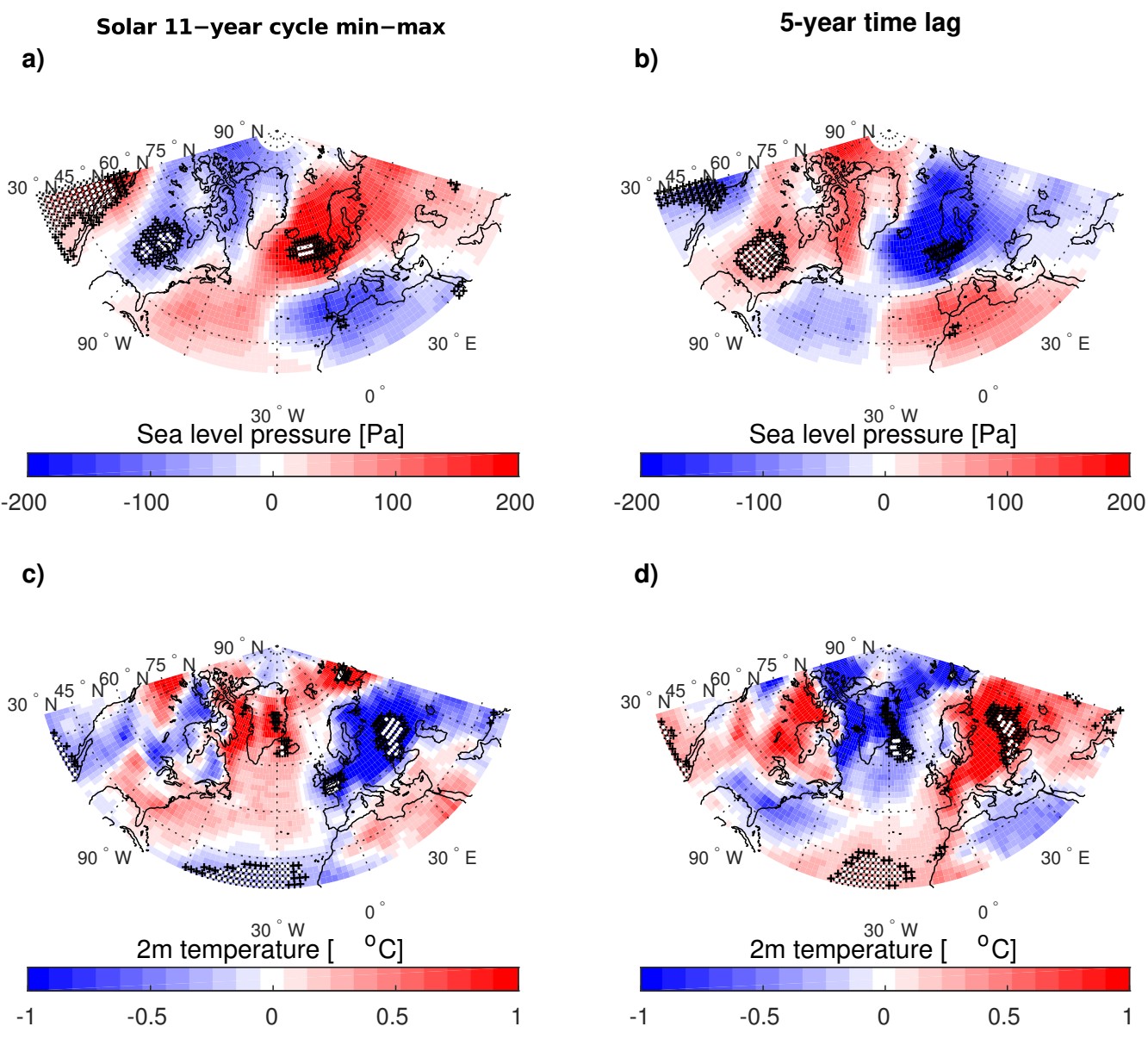

**Figure 6.** 20CR (1948-2010) atmospheric response to solar forcing. **a** DJF SLP anomalies [Pa] in response to the 11-year solar cycle (solar min. minus solar max. defined in Figure S12). **b** DJF SLP anomalies [Pa] in response to the 5-year lagged 11-year solar cycle (solar min. minus solar max.). **c, d** corresponding figures to **a, b**, but for T2m [$^o$C]. The white stippling indicates significant anomalies $p < 0.05$, and black stippling indicates significant anomalies $p < 0.1$ (two-tailed Student's $t$-test). The time interval for this analysis in limited to 1948-2010 due to limitation of the data quality prior to this, although similar results can be achieved for the period 1851-2010.

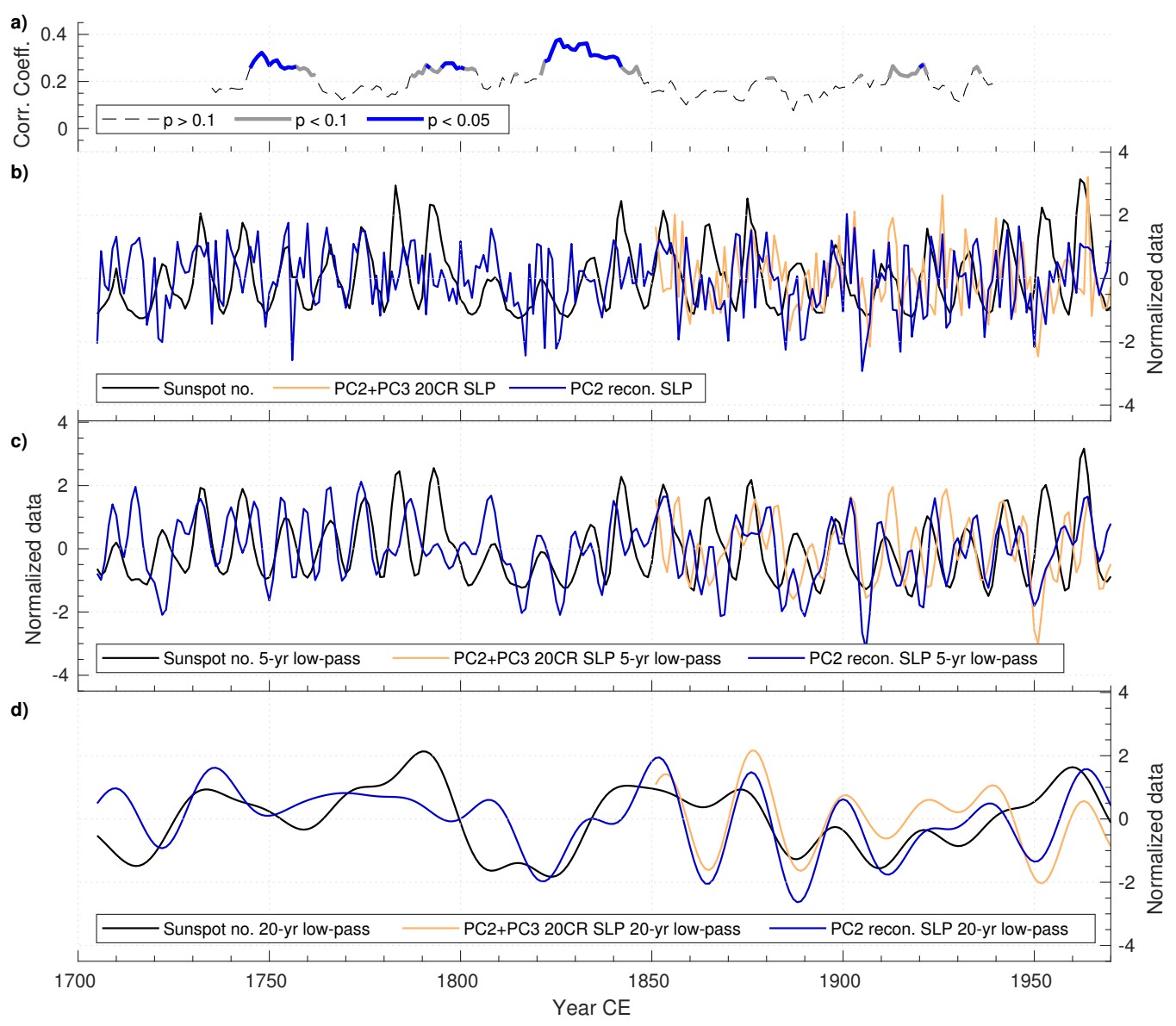

**Figure 7.** Reconstructed PC2 of SLP plotted with the 5-year lagged sunspot number. **a** Moving 61-point correlation between reconstructed PC2 of SLP and sunspot number shifted for a 5-year time lag. **b** Time series of sunspot number shifted for a 5-year time lag, PC2+PC3 of 20CR SLP (see text and Table 4) and PC2 of reconstructed SLP. **c** Same as **b**, except filtered with a 5-year low-pass filter. **d** Same as **b**, except filtered with a 20-year low-pass filter.