# Peer review of "Solar and volcanic forcing of North Atlantic climate inferred from a process-based reconstruction"

_Climate of the Past, 2018_

## Referee Comment (RC1) · Anonymous Referee #1 · 10 May 2018

The authors in this interesting study fuse measurements of isotopic composition of oxygen with simulated results from an isotope enabled global climate model to reconstruct seal level pressure and temperature changes over the 1241-1970 period. The model gives spatial maps of pressure and temperature for each year, in an ensemble way, which is raked according to minimization of the distance between the simulated and measured PCs of the oxygen isotope. Then the authors infer volcanic and solar influences in the reconstruction. The propoced method is novel and may return some useful information about external forcings over the last millennium. The manuscript is generally well written and figures are presented in a crystal way. In the following, I am listing some thoughts/comments which hopefully will increase clarity of the manuscript.

[Figure]

Major comments In my view, this study is more about evaluating the proposed methodology rather than detecting volcanic and solar signals. For example, the length of the text dedicated to volcanic effects in the reconstruction is a single paragraph only (1st paragraph page 7). Given the extended length of the detected response compared to other studies I would strongly encourage authors to provide more details. For example, to what extend results are influenced by the performance of ECHAM to eruptions. How the response looks if picked up actual model years not reconstructed? An additional validation method would be to examine the sensitivity of the results to the model choice. The PMIP3 simulations are ideal to this this purpose but I admit is a lot of additional effort.

Added value of reconstruction The added value of the reconstruction is not properly assessed. The reader is left wondering whether similar skills (e.g. correlations) could be obtained by the model simulation alone. The authors should at least mention the how much correlations between the modelled NAO and 20CR or other reconstructions are improved in the reconstruction. Likewise for the modelled PC2.

Use of solar forcing There is an extensive description of the TSI forcing used (is there SSI variability? ) in the model which differs from those recommended in the PMIP3. I guess the continuity of the Muscheler reconstruction throughout the last 500 or so years does not immediately make it superior over the other reconstructions. I am wondering however if the choice of TSI really matters, given that the reconstruction does not really benefit by the timing of TSI forcing (authors comment on that in p.5 l6). TSI and other forcings are just increasing the phase space of the surface patterns which may or may not be chosen eventually in the reconstruction. How different the reconstruction would be by feeding in model data from a constant forcing (control) simulation?

Top-down mechanism is missing (related to the previous comment) I am pretty sure the ECHAM5 version does not properly resolves the so called "top-down" mechanism of SSI forcing on the stratosphere and subsequence changes in the troposphere and surface. To my understanding this is a key component of the proposed solar-NAO link.

[Figure]

Given that weakens of the model it is not surprising that little evidence is found between NAO and solar forcing. So, the argument that solar forcing has little effect on NAO is not well justified. Likewise, in the discussion (p.8 last paragraph), the argument about increasing blocking comes hand in hand with the stratospheric response and the "top-down" mechanism, with is missing here. So it is likely that you may get right (to some extent) responses from wrong reasons.

Solar responses and time lags The 11-yr solar cycle SLP anomalies in Figure 4 do not seem to support the results of Gray et al., 2013. It is the different period that change signals? Could you please calculate signals from 1850 as to be consistent with the HadSLP2 years? The argument that signals maximize with 5 years lag is strange. 20CR shows almost opposite signals in lag 5 compared to lag 0, an antisymmetry not obvious in the reconstruction (Figure 4). My guess is that the correlation of PC2 with the 11-yr happened to maximize at lag 5. Thinking linearly, I would expect a negative correlation at lead time +1 years or lag 0, but in table 2 I see just zero. Even at lag 5 the explained variance by the 11-yr solar forcing is tiny. Authors should comment on the amazing correlation between reconstructed PC2 and 20CR PC2+PC3 (figure 6c). Why is it so?

Other It is a bit strange to present anomalies in solar minima minus maxima. Most of the studies I am aware of show max-min. Please describe the method inferring significance. It is mentioned in figures but should be clearly described in the manuscript. Same for the type of filtering applied on PC2s.

P.3 L.16 time lag of 4-6 years: This is solar maximum, no? (see my comment above) P.3 l.23 "" fully prescribed CO2": what do you mean? How it differs in E1-COSMOS?

p.3 parag.2.1: What is the horizontal resolution of the model. How many layers in the stratosphere? p.5. l.1: "fitting the PCs of...": please make it clear what is the criterion. The sum of PCs? Individual PCs? p.5 l.16: to me it seems that only DYE-3 shows high correlation. In other high accumulation sites correlation is much lower.

---

## Referee Comment (RC2) · Anonymous Referee #2 · 17 May 2018

——————— General comments ———————

This paper is proposing a new methodology for reconstruction atmospheric variability modes over the last millennium. This technique is using a climate model simulation including the online computation of the variation in concentration of oxygen isotopes, which are then compared with ice cores records. This comparison is then allowing, through a statistical approach, to find the winter atmospheric circulation that fits the best with the observations. Using this approach, the authors propose a new reconstruction of the two first modes of the variability of the atmospheric circulation over the North Atlantic sector. Then, the authors compare their reconstruction with time series

of volcanic eruptions and solar forcing and find a clear signature of the former on the first mode, and a smaller signature of the latter on the second mode.

This is a very interesting study, proposing an innovative approach to reconstruct past climatic modes of variability. The analysis led is thorough and the results found are very impressive. As compared to a former reconstruction of the NAO (first mode of atmospheric circulation variability in winter), the new one exhibits slightly better scores (not sure the difference is really significant) of validation and is also showing no loss of variance when coming back in time. The study is also providing a reconstruction of the second mode of variability, usually denominated as the East Atlantic Pattern (EAP). These are very useful reconstructions for the climate community, and the manuscript clearly deserves publication.

I have noticed quite a number of issues mainly related with the clarity of the explanations, and also, from time to time, I have noticed some too enthusiastic evaluation of their work by the author, while a scientific work requires objectivity and discussion of strengths but also of potential weaknesses and caveats of the methods, which are not sufficiently discussed at the moment.

In particular, I think it would be worth discussing:

1) The fact that the model used is quite coarse resolution, so that it may have large biases, potentially even in its representation of the atmospheric circulation and of its variability modes, which need to be better depicted.

2) The reconstruction is finally relying on very few data, only 8 cores from Greenland, while the Atlantic sector is huge. Validation score are relatively high (even though a correlation of 0.5 stands for less than 30% of the variance explained...), but we can easily envisage that more data will be helpful to improve the reconstruction. A discussion of this will be useful I think.

3) The link with solar forcing is far from being straightforward, and the authors push their

conclusions a bit too far I should say. Also, the variability is not necessarily related with external forcing, and large amount of variance can be purely stochastic (i.e. internal to the climate system). While lots of research is led on this topic, notably to explain the little ice age, this is simply not discussed at all here. What is the percentage of variance in our reconstruction that is not related to any external forcing? This is not easy to isolate, but a rough estimate would be interesting (e.g. Wang et al. 2017).

With a proper discussion of the following points, I think the manuscript will clearly deserve to be published in climate of the past.

———————— Specific points: ————————

- p.1, l.1: the two first sentences can be interpreted as almost contradictory. Can you please reformulate them to clarify what you have in mind here?

- P.1, l. 3: add "of" before "the effects"

- P. 1, l. 3: "A positive phase...": This is not true. Some of the historical eruptions have been followed by negative phase of the NAO (e.g. Agung) and the link between NAO and volcanic eruptions over the historical era is very small and sensitive to the selected eruptions (cf. Swingedouw et al. 2017, cited in the ms.). Please clarify.

- P. 1, l. 10: you should specify here that this is for winter season.

- P. 1, l. 16: "we observe a similar response" is not clear at all. Please reformulate what you mean here. The next sentence concerning "blocking frequency" is also quite unclear. Please avoid the word "likely" as it has a very specific meaning in IPCC report, which is not the one used here, since I have not seen any proper analysis of changes in blocking frequency in the manuscript.

- P. 1, l. 19: You'll need to define what you mean here by little ice age in terms of time period.

- P. 1, l. 19: "a clear link" is quite subjective sentence. Can you be more specific (i.e.

quantitative: the correlation is quite small so that the link is not that clear I would say).

- P. 1, l. 20: no "s" to "pattern"

- P. 2, l. 1-2: It could be worth to specify here winter season, since the whole paper is focused on this season.

- P. 2, l. 6-7: "Past changes. . ..": Variability in atmospheric circulation is usually believed to be mainly stochastic, with a weak imprint of external forcing. This is why seasonal or decadal prediction are so difficult: the atmosphere is dominated by chaos, with very small part of the variance that is predictable. . . This is in contradiction with the word "attributed" from your sentence. Please clarify. Also, please add some references to substantiate the claim from this sentence.

- P. 2, l. 10: add coma after "millennium"

- P. 2, l. 14-15: "The NAO has also. . .": This sentence is very confusing. Please rephrase by just saying that solar variations are slightly correlated with SLP over the historical era in a few areas.

- P. 2, l. 23-24: what is also very important is the amplitude of the influence i.e. the variance explained by solar variability. It is possible this forcing has a slight influence, but how does it compare to the noise, i.e. the purely stochastic variation of the NAO. The signal-to-noise ratio is indeed key to evaluate here.

- P. 3, l. 1: replace "particularly" by "particular".

- P. 3, l. 21-25: can you please provide a few more information on this model like its resolution? A brief description of its biases in the region analyzed will be also enlightening to evaluate the limit of this model for the exercise performed. How does the isotopic variations in the ice core sites compare with observations? How many grid points are covering Greenland?

- P. 4, l. 11: this section 2.3 is key to the paper. Nevertheless, I find it a bit difficult

to follow. They are very few references that support the method presented, so that it seems that this approach is entirely new. Is that correct? If so, I would like to have a longer description of it and a few examples to fully understand how the method works. Furthermore, I'm wondering how sensitive the results to the X2 measure are, which sounds a bit arbitrary and not very well justified.

- p. 5, l. 2: The authors depict "39 time series". This is not crystal clear what the time is here and what the members are. Can you please be more specific to improve this description (a scheme could be useful as well for instance).

- P. 5, l. 2: "reshuffled model": a model is a tool with which you can perform simulations, providing climate variables. Thus, I'm not sure that "reshuffled model" is a proper terminology. I think, you are speaking here of the "reshuffled output from the simulation". This remark is true throughout the manuscript where model is used inconsistently.

- P. 5, l. 29: "Note...": This is not a very expression. Normally all what is written in the manuscript is worth to be noted.

- P. 5, l. 30: "is associated": it is not clear what is substantiating this claim. Please clarify.

-P. 5, l. 30: "Figure S3": I find the pattern of EOF2 very different between model simulation and reanalysis. This should be said somewhere.

- P. 5, l. 33: add a coma after "reconstructions"

- P. 6, l. 10: what about the biases of the model? I believe this can also explain your differences between model simulations and reanalysis! Climate models are not providing perfect representation of reality, the opposite is true.

- P. 6, l. 14 add "in" after "consists"

- P. 7, l. 16: "Scandinavian blocking-type pattern". You should support this by a reference. Also, I would have argued that a Scandinavian blocking is referring to a change

in frequency and is usually an anticyclone that remains blocked over the Scandinavia, following the weather regime approach (e.g. Vautard et al. 1990, Ortega et al. 2014 in line with ice core analysis and weather signature). Can you please clarify this sentence?

- P. 7, l. 22: "it should be noted". Avoid this type of subjective comment.

- P. 7, l. 30: "looks slightly different": this is a very subjective judgments, and I would rather say that they have hardly anything in common. Can you please provide a more objective metric of their similarity (spatial correlation for instance)?

- P. 7, l. 31: "wave structure". OK, but the signs are almost opposite in Fig. 4 a and c... So, this is not very convincing as a similar pattern!

- P. 8, l. 3: maybe state that this secondary pattern is usually denominated as the EAP. Also, it is better to avoid "likely", except when the meaning is in line with IPCC precise definition.

- P. 8, l. 14: this positive feedback is very weak, and hardly significant in the observations... (which is why the NAO is so difficult to predict at the seasonal scale, and the recent improvements seem to be more related with tropical teleconnection and frontal dynamics around the Gulf Stream region...).

- P. 8, l. 30: this line is very speculative and not supported by any references. The influence of the North Atlantic SST on atmospheric circulation is mainly not significant in winter season in the observation, or at least largely debated, so this hand-made explanation sounds a bit speculative I should say. I would at least replace "likely" l. 28 by "possible" given its high level of speculation.

- P. 9, l. 1: "Although the authors..." Indeed, they do not... They rather propose that the LIA would have been intrinsic and related to (unforced) rapid changes of the SPG. Furthermore, detection-attribution analysis (e.g. Schurer et al. 2014) found almost no signature of solar forcing on NH temperature, implying that LIA was hardy forced by

solar variations. You should discuss this as well I would say. How do you reconcile this with your interpretations?

- Figure 1: what about adding a sliding window correlation here? It could also be useful for Figure 2 and Figure 6. It will help to evaluate when the time series are well-correlated by another mean than just by eye.

References:

Ortega P., D. Swingedouw, V. Masson-Delmotte, C. Risi, B. Vinther, P. Yiou, R. Vautard, K. Yoshimura (2014) Characterizing atmospheric circulation signals in Greenland ice cores: insights from the weather regime approach. Climate Dynamics 43 (9-10), pp. 2585-2605, DOI 10.1007/s00382-014-2074-z. Schurer A.P., Tett S.F.B., Hegerl G.C., 2014. Small influence of solar variability on climate over the past millennium. Nat. Geosci., 7, 104-108. Vautard R (1990) Multiple weather regimes over the North Atlantic analysis of precursors and successors. Mon Weather Rev 118:2056–2081 Wang et al. (2017) Internal and external forcing of multidecadal Atlantic climate variability over the past 1,200 years• Nature Geoscience volume 10, pages 512–517.

---

## Author Comment (AC1) · 20 Jun 2018

Below we list point-by-point responses to all reviewer comments. We are grateful for all the comments by the reviewers, which has helped us clarify and improve this manuscript. Our replies are in *italics* and all changes to the text are in *blue*.

Referee #1

The authors in this interesting study fuse measurements of isotopic composition of oxygen with simulated results from an isotope enabled global climate model to reconstruct seal level pressure and temperature changes over the 1241-1970 period. The model gives spatial maps of pressure and temperature for each year, in an ensemble way, which is raked according to minimization of the distance between the simulated and measured PCs of the oxygen isotope. Then the authors infer volcanic and solar influences in the reconstruction. The propoced method is novel and may return some useful information about external forcings over the last millennium. The manuscript is generally well written and figures are presented in a crystal way. In the following, I am listing some thoughts/comments which hopefully will increase clarity of the manuscript.

*Thank you for the positive review, and the useful comments which has helped us to clarify our manuscript.*

Major comments In my view, this study is more about evaluating the proposed methodology rather than detecting volcanic and solar signals. For example, the length of the text dedicated to volcanic effects in the reconstruction is a single paragraph only (1st paragraph page 7). Given the extended length of the detected response compared to other studies I would strongly encourage authors to provide more details. For example, to what extend results are influenced by the performance of ECHAM to eruptions. How the response looks if picked up actual model years not reconstructed? An additional validation method would be to examine the sensitivity of the results to the model choice. The PMIP3 simulations are ideal to this this purpose but I admit is a lot of additional effort.

*While it is true that we spend quite some space in the manuscript on evaluating the method, we believe that this is necessary, since this is the first paper presenting this method. With respect to the response to the volcanic forcing: As written line 2-3, page 5 of the manuscript "The temporal succession of the reshuffled model fits do not resemble the order of the years in the original model [ed: run]...". This is also true for the timing of the volcanic and solar forcing used for the model simulation. This means that the response of the ECHAM5/MPI-OM model to volcanic forcing is not directly relevant for the reconstructed response to volcanic eruptions. We have clarified this in the Methods text. For reference, the response to volcanic forcing of the ECHAM5/MPI-OM model has been studies extensively (e.g. Zanchettin et al. 2012, Guðlaugsdóttir et al., accepted).*
*The reviewer suggests that our method is tested using the PMIP3 simulation. It would indeed be desirable with such a test, however this would require the PMIP3 model suite first to have isotope diagnostics implemented in the code. Swingedouw et al. (2017) (cited in the Introduction) reviewed the PMIP3 models' NAO response to major tropical eruption of the past millennium of the PMIP3 models and found a large scatter in the models' response, with little coherence with the reconstructed NAO response by Ortega et al. (2015). Our reconstructed response to volcanic forcing is fairly straight forward to describe, and we feel the result it self does not benefit further from an extended description. However, following the reviewers suggestion we have added more details in the Discussion section concerning these questions.*

*Inserted, page 5, line 8: "Since the timing of the forcings used for the model simulation is not a factor for the reconstruction, it also means that the model performance for the response to forcings does not influence the reconstruction."*

*Inserted, page 8, line 6: "As shown in the comparison of the response to volcanic forcing between our reconstruction and the MC reconstruction by Ortega et al. (2015) (Figure 3 c), the mean response look qualitatively very similar in the two reconstructions. However, as already mentioned, due to the preserved high frequency variability our reconstruction shows both a more immediate and persistent NAO response to volcanoes. This underlines the importance of producing climate reconstructions that do preserve high frequency variability, in particular if the reconstruction is used as baseline for model evaluation. In our analysis of the reconstructed response to volcanic eruptions we choose eruptions larger than or of similar magnitude as the 1991 Pinatubo eruption ($< -6\ Wm^2$). As discussed by Swingedouw et al. (2017) climate effects of smaller eruptions can be difficult to detect due to stochastic climate variability. We find that we can detect the an impact on NAO from tropical eruptions selected in the range from -4 $Wm^2$ to -8 $Wm^2$ (Sigl et al., 2015), yielding a significant positive NAO one year after the eruptions, on average. This appears to be the limit of detection for our reconstruction, possibly owing both to the partly stochastic variability of the NAO and to noise in the reconstruction."*

Added value of reconstruction The added value of the reconstruction is not properly assessed. The reader is left wondering whether similar skills (e.g. correlations) could be obtained by the model simulation alone. The authors should at least mention the how much correlations between the modelled NAO and 20CR or other reconstructions are improved in the reconstruction. Likewise for the modelled PC2.

*The variability of the NAO is to a large extent driven by internal variability, although a proportion is driven by external forcing and interaction with the ocean (Hurrell et al., 2003). Running an atmospheric model with SSTs prescribed from observations, as well as observed aerosols and stratospheric forcing yields a correlation to observed NAO of only about 0.25 (Compo et al., 2011). This is the reason for the many efforts to reconstruct the NAO, e.g. as reviewed by Pinto and Raible (2012), and part of the motivation of our study (page 2, line 24-30).*
*We compare with the two NAO reconstructions by Ortega et al. (2015) as well as the NAO reconstruction Luterbacher et al. (2004) in Table 1. We also compare the skill in relation to observed NAO (Jones et al., 1997) of our NAO reconstruction to that of the skill of Ortega et al. (2015), where our reconstruction shows slightly higher skill, and with much improved year-to-year variability (page 6 , line 27-35). We are no aware of any other published reconstruction of PC2 of SLP that we might compare with.*
*In the revised manuscript we now highlight also the comparison the NAO reconstruction by Luterbacher et al. (2004), which was not discussed in the text before.*

*Reformulated, page 6, line 21: "For an independent comparison we use reconstructed NAO and gridded of SLP (Luterbacher et al., 2001), and reconstructed temperature (Luterbacher et al., 2004) over Europe."*

*Inserted, page 6, line 24: "For the NAO reconstruction by Luterbacher et al. (2001) our reconstruction shows slightly lower correlation on interannnual time-scales compared to the correlation to NAOmc, but similar correlation on decadal to multi-decadal time-scales (Table 1)."*

Use of solar forcing There is an extensive description of the TSI forcing used (is there SSI variability? ) in the model which differs from those recommended in the PMIP3. I guess the continuity of the Muscheler reconstruction throughout the last 500 or so years does not immediately make it superior over the other reconstructions. I am wondering

however if the choice of TSI really matters, given that the reconstruction does not really benefit by the timing of TSI forcing (authors comment on that in p.5 l6). TSI and other forcings are just increasing the phase space of the surface patterns which may or may not be chosen eventually in the reconstruction. How different the reconstruction would be by feeding in model data from a constant forcing (control) simulation?

*The TSI reconstruction is described in detail here since it, as mentioned by the reviewer, differs slightly from the PMIP3 reconstruction, but does however agree well with the updated PMIP4 forcing. Please note that the model run was performed before the PMIP4 forcing was available. This is a TSI forcing only, which ECHAM5 divides in 4 bands (1 visible+UV and 3 near infra red) (Roeckner et al., 2003). As the reviewer points out the exact forcing of the model is not important for the reconstruction. Although we find that the timing of the forcings driving the model simulation are not crucial for the reconstruction, and there is no preference to forced versus unforced model years, the combined effect of all the forcings increases the model variance, which produces a more diverse sampling space for the method to pick from. We have clarified the role of model forcings in the methods section (see reply to previous comment), and added a paragraph in the Discussion on the role of the model run.*
*However, for the analysis of the reconstructed longterm solar response we also use this TSI reconstruction, and here the timing and amplitude of the solar cycles are important. In this case the timing is crucial and our solar activity reconstruction is based on $^{14}C$ data that should not contain any dating uncertainties.*

*Inserted, page 8, line 6:* *"The model simulation used for our reconstructed translates the climate variability recorded in the Greenland ice cores to climate variability in the North Atlantic region. In the initial test of the isotope variability it is shown, that the spatio-temporal $\delta^{18}O$ variability of the ice cores is well represented by the model (Figure S6). This is a fundamental prerequisite that allows us to match the modeled $\delta^{18}O$ to the ice core $\delta^{18}O$ year-by-year. While the skill of the reconstruction is higher in the vicinity around Greenland, the reconstruction shows significant correlations to reanalysis data wide spread across the North Atlantic region. This skill depends on i) the integrative nature of the $\delta^{18}O$ as recorded in the ice cores, and represented by the modeled $\delta^{18}O$ ii) the modeled atmospheric teleconnection patterns in terms of temperature and circulation, and iii) how these patterns are connected to modeled $\delta^{18}O$ for Greenland. Clearly, the reconstruction is strongly dependent on the climate model when it comes to whether or not it is possible at all to use our method, and when it comes to the skill of reconstructed spatial patterns. The resolution of our model model simulation is relatively course and using a higher resolution simulation could improve the representation of several processes. For example, vapor transport to dry polar regions is often inhibited in models with courser resolution, resulting in too little precipitation in the interior of ice sheets and a positive bias in $\delta^{18}O$ (Masson-Delmotte et al., 2008, Sjolte et al., 2011). This is related to cloud parameterizations and course resolution models having difficulties in explicitly representing frontal zones in connection with synoptic weather systems. The orography in course resolution models is also more smooth, loosing orographical features such as the southern dome of the Greenland ice sheet, which also affects the circulation and small scale spatial variability. In our approach we match the modeled PCs of $\delta^{18}O$, meaning that we are matching regional scale patterns in $\delta^{18}O$, which partly addresses the problem of matching course model output to site specific proxy data. However, having a higher resolution model simulation could for example improve the spatio-temporal representation of Greenland $\delta^{18}O$, allowing more than 3 PCs to be fitted, and generally giving a better representation of temperature, pressure and precipitation in the reconstruction. For reasons discussed above, it would be desirable using different GCMs to test for model dependencies of the reconstruction, as well as testing for added value of ensemble reconstructions with several different GCMs. Doing these tests is presently limited by the availability of millennium length simulations using isotope enabled GCMs."*

Top-down mechanism is missing (related to the previous comment) I am pretty sure the ECHAM5 version does not properly resolves the so called "top-down" mechanism of SSI forcing on the stratosphere and subsequence changes in the troposphere and surface. To my understanding this is a key component of the proposed solar-NAO link. Given that weakens of the model it is not surprising that little evidence is found between NAO and solar forcing. So, the argument that solar forcing has little effect on NAO is not well justified. Likewise, in the discussion (p.8 last paragraph), the argument about increasing blocking comes hand in hand with the stratospheric response and the "top-down" mechanism, with is missing here. So it is likely that you may get right (to some extent) responses from wrong reasons.

*The reviewer is right in that the top-down hypothesis for solar influence of climate has been put forward in several studies, and the solar influence on the stratosphere is well-documented. However, as the model run is reshuffled (based on real-world isotope data), the reconstruction largely depends on the isotope-climate relation in the model and how this is matched to the ice core variability (see also replies to comments above). This means that whether or not the model has a realistic representation of stratosphere-troposphere coupling is less important, provided that the model results give a realistic ensemble of possible tropospheric circulation regimes. Given the limited representation of the stratosphere in the 19-layer version of the ECHAM5 model it is also unlikely that we can investigate a possible top-down mechanism from the reconstruction alone.*

Solar responses and time lags The 11-yr solar cycle SLP anomalies in Figure 4 do not seem to support the results of Gray et al., 2013. It is the different period that change signals? Could you please calculate signals from 1850 as to be consistent with the HadSLP2 years? The argument that signals maximize with 5 years lag is strange. 20CR shows almost opposite signals in lag 5 compared to lag 0, an antisymmetry not obvious in the reconstruction (Figure 4). My guess is that the correlation of PC2 with the 11-yr happened to maximize at lag 5. Thinking linearly, I would expect a negative correlation at lead time +1 years or lag 0, but in table 2 I see just zero. Even at lag 5 the explained variance by the 11-yr solar forcing is tiny. Authors should comment on the amazing correlation between reconstructed PC2 and 20CR PC2+PC3 (figure 6c). Why is it so?

*As we see it, the response seen in Gray et al. (2013) Figure 1, could be consistent with our results. With no lag, their results show negative correlations across the British Ilse and Scandinavia (although not significant), which is consistent to the high-pressure anomaly we see in the reconstructed response (anomalies for solar min-max, so the anomalies are reverse compared to the regression in Gray et al.). Gray et al. also see a stronger, and reversed, response at 5-year lag, which is significant. There are a number of differences between the study of Gray et al. (2013, 2016) and our study i) Gray et al. uses a regression model, while we pick low vs. high solar activity ii) the time period chosen by Gray et al. is 1870-2010, and iii) Gray et al. uses HadSLP2 data. The analysis of the solar response is sensitive to the method, the time period chosen, and the climate data set. Since our reconstruction only goes to 1970, we cannot perform a consistent comparison to Gray et al., and if we chose a short period, e.g. 1870-1970, instead of 1700-1970, the response is qualitatively similar but weaker and not significant in the reconstruction. The noise level in any reconstruction can be expected to be higher than in observations, and the solar influence in climate is difficult to identify even in observations. So we rely on the long-term average response from solar cycle to solar cycle to average out noise and get a mean response. Comparing the areas of significant response in Figure 4 and Figure 5, we find the reconstructed response to the 11-yr cycle, with and without time lag to be consistent with the response seen in the 20th Century Reanalysis. The analysis shows strongest correlation with 5-year time lag between PC2 and the sunspot number, but correlations are significant with 3-7 year time lag, meaning that this 5-year lag should*

*not be taken as an absolute number, but the average maximum lag correlation. We do actually see a weak negative correlation between PC2 and the sunspot number with no time lag, which is also evident when looking at the time series. We have now added a version of figure 6 with no time lag in the supplementary (see figure below), which illustrates the weaker and more inconsistent correlation. In particular when looking at the moving window correlation, which we added on request by Referee # 2.*

*We agree with the reviewers comment that the variance explained by solar forcing is small. However, we do not think that this makes our conclusions less important. It is known that large part of climate variability is driven by internal processes and are hence unforced. Thus, we cannot expect high correlation coefficients, although the correlations between reconstructed PC2 and solar forcing at decadal ($r = 29$, $p < 0.01$) and centennial time scales ($r = 0.6$, $p < 0.01$) are not negligible. It is important to identify and understand the forced variability in the climate system, as this allows for a better understanding of past climate, and a better skill in predicting future climate. Page 6, line 4-10 we discuss the correlation between reconstructed PC2 and 20CR PC2+PC3. We have clarified this in the revision.*

*Inserted, page 6, line 9: "This indicates that the variability projected on PC2 and PC3 of the reanalysis data is partly summarized in PC2 of the reconstruction."*

[Figure]

*New figure in revised supplementary. Reconstructed PC2 of SLP plotted with the sunspot number (no time lag). **a** Moving 61-point correlation between reconstructed PC2 of SLP and sunspot number. **b** Time series of sunspot number, PC2+PC3 of 20CR SLP (see text and Table 2) and PC2 of reconstructed SLP. **c** Same as **b**, except filtered with a 5-year low-pass filter. **d** Same as **b**, except filtered with a 20-year low-pass filter.*

Other It is a bit strange to present anomalies in solar minima minus maxima. Most of the studies I am aware of show max-min. Please describe the method inferring significance. It is mentioned in figures but should be clearly described in the manuscript. Same for the type of filtering applied on PC2s.

*We show min-max because we are interested in the response to solar minimum context of the discussion of the role of solar forcing during the Little Ice Age. Hence, we think it is most convenient for the reader to see the response to solar minimum as also done in similar studies (Adolphi et al., 2014; Ineson et al., 2011; Martin-Puertas et al., 2012).*
*We have added a paragraph in the Methods section describing methods used for significance, and filtering in the manuscript.*

P.3 L.16 time lag of 4-6 years: This is solar maximum, no? (see my comment above)

*This is referring to the time lag response with this analysis done on PC2, using the time series of PC2 and sunspot number, so there is no choice of solar maximum or minimum in the analysis. This has been clarified in the revision.*

*Reformulated, page 3, line 16:* *"We achieve the strongest correspondence between the solar forcing and reconstructed PC2 of SLP with a time lag of 5 years, indicating that an ocean-atmosphere feedback is in play."*

P.3 l.23 "" fully prescribed CO2": what do you mean? How it differs in E1-COSMOS?

*For the E1-COSMOS ensemble the model is configured with a carbon cycle module which drives part of the variability in the atmospheric CO2. We have specified this difference between the E1 ensemble and our run in the revised version of the manuscript.*

p.3 parag.2.1: What is the horizontal resolution of the model. How many layers in the stratosphere?

*The atmospheric component of the model, ECHAM5, is a spectral model and we ran it in T31 resolution, corresponding to $3.75^o$ x $3.75^o$ (lon, lat), with 19 vertical layers. 5 of these layers are in the stratosphere. We have included this information in the revised version of the manuscript.*

p.5. l.1: "fitting the PCs of. . .": please make it clear what is the criterion.
The sum of PCs? Individual PCs?

*This is referring to the time series of PCs of the ensemble members evaluated using Eq. 1. We have reformulated this part in the revision.*

*Reformulated, page 4, line 32 to page 5 line 1:* *"We define the sorted model output as ensemble members, such that the best fitting model year, for each year of the ice core data, belongs to ensemble member 1, the second best fitting model years belong to ensemble member 2, and so forth. Using a Chi-square goodness-of-fit test, with respect to the measure in Eq. 1, we evaluate the ensemble members against the PCs of Greenland ice core $\delta^{18}O$ and reject model fits with likelihood $p > 0.01$ of not fitting the ice core data. This leaves us with 39 time series of reshuffled model data fitted to the ice core data."*

p.5 l.16: to me it seems that only DYE-3 shows high correlation. In other high accumulation sites correlation is much lower.

*Indeed, high accumulation and multiple cores from the same location help constrain the noise. This appears to be the deciding factors. We have reformulated this in the revision.*

Referee #2

[Figure]

─────────── General comments ───────────
This paper is proposing a new methodology for reconstruction atmospheric variability
modes over the last millennium. This technique is using a climate model simulation
including the online computation of the variation in concentration of oxygen isotopes,
which are then compared with ice cores records. This comparison is then allowing,
through a statistical approach, to find the winter atmospheric circulation that fits the
best with the observations. Using this approach, the authors propose a new recon-
struction of the two first modes of the variability of the atmospheric circulation over the
North Atlantic sector. Then, the authors compare their reconstruction with time series of volcanic
eruptions and solar forcing and find a clear signature of the former on the
first mode, and a smaller signature of the latter on the second mode.

This is a very interesting study, proposing an innovative approach to reconstruct past
climatic modes of variability. The analysis led is thorough and the results found are
very impressive. As compared to a former reconstruction of the NAO (first mode of
atmospheric circulation variability in winter), the new one exhibits slightly better scores
(not sure the difference is really significant) of validation and is also showing no loss
of variance when coming back in time. The study is also providing a reconstruction of
the second mode of variability, usually denominated as the East Atlantic Pattern (EAP).
These are very useful reconstructions for the climate community, and the manuscript
clearly deserves publication.

*We thank the reviewer for the positive review and the many constructive comments and suggestions.*

I have noticed quite a number of issues mainly related with the clarity of the explana-
tions, and also, from time to time, I have noticed some too enthusiastic evaluation of
their work by the author, while a scientific work requires objectivity and discussion of
strengths but also of potential weaknesses and caveats of the methods, which are not
sufficiently discussed at the moment.

In particular, I think it would be worth discussing:

1) The fact that the model used is quite coarse resolution, so that it may have large
biases, potentially even in its representation of the atmospheric circulation and of its
variability modes, which need to be better depicted.

*We have added a paragraph in the discussion about the role of the climate model in the
reconstruction.*

*Inserted, page 8, line 6: "The model simulation used for our reconstructed translates the climate
variability recorded in the Greenland ice cores to climate variability in the North Atlantic region. In
the initial test of the isotope variability it is shown, that the spatio-temporal $\delta^{18}O$ variability of the
ice cores is well represented by the model (Figure S6). This is a fundamental prerequisite that
allows us to match the modeled $\delta^{18}O$ to the ice core $\delta^{18}O$ year-by-year. While the skill of the
reconstruction is higher in the vicinity around Greenland, the reconstruction shows significant
correlations to reanalysis data wide spread across the North Atlantic region. This skill depends on
i) the integrative nature of the $\delta^{18}O$ as recorded in the ice cores, and represented by the modeled
$\delta^{18}O$ ii) the modeled atmospheric teleconnection patterns in terms of temperature and circulation,
and iii) how these patterns are connected to modeled $\delta^{18}O$ for Greenland. Clearly, the
reconstruction is strongly dependent on the climate model when it comes to whether or not it is*

*possible at all to use our method, and when it comes to the skill of reconstructed spatial patterns. The resolution of our model model simulation is relatively course and using a higher resolution simulation could improve the representation of several processes. For example, vapor transport to dry polar regions is often inhibited in models with courser resolution, resulting in too little precipitation in the interior of ice sheets and a positive bias in $\delta^{18}O$ (Masson-Delmotte et al., 2008, Sjolte et al., 2011). This is related to cloud parameterizations and course resolution models having difficulties in explicitly representing frontal zones in connection with synoptic weather systems. The orography in course resolution models is also more smooth, loosing orographical features such as the southern dome of the Greenland ice sheet, which also affects the circulation and small scale spatial variability. In our approach we match the modeled PCs of $\delta^{18}O$, meaning that we are matching regional scale patterns in $\delta^{18}O$, which partly addresses the problem of matching course model output to site specific proxy data. However, having a higher resolution model simulation could for example improve the spatio-temporal representation of Greenland $\delta^{18}O$, allowing more than 3 PCs to be fitted, and generally giving a better representation of temperature, pressure and precipitation in the reconstruction. For reasons discussed above, it would be desirable using different GCMs to test for model dependencies of the reconstruction, as well as testing for added value of ensemble reconstructions with several different GCMs. Doing these tests is presently limited by the availability of millennium length simulations using isotope enabled GCMs."*

2) The reconstruction is finally relying on very few data, only 8 cores from Greenland, while the Atlantic sector is huge. Validation score are relatively high (even though a correlation of 0.5 stands for less than 30% of the variance explained. . .), but we can easily envisage that more data will be helpful to improve the reconstruction. A discussion of this will be useful I think.

*We have added a paragraph in the discussion about the selection of proxy records, number of records and quality of records (dating, resolution, ...).*

*Inserted, page 8, line 6 (after model discussion): "We selected the proxy records for this study based on the criterion of having seasonal resolution, small dating uncertainty, a long time span and a wide regional spread. In order to provide a quantitative link to the isotope enabled GCM we selected only isotope-based proxies. For the time being, this leaves us with the 8 Greenland ice cores used in this study. Other seasonal resolution ice cores from Greenland are available, but only covering a limited time span, and comparing to these cores shows that the reconstructed $\delta^{18}O$ also has good correlation to these cores (Figure S2). However, including more Greenland ice cores of similar quality would generally improve the signal to noise ratio of the reconstruction, and such records should be included if available for subsequent studies. Obtaining seasonal resolution in ice core data is mainly limited by the accumulation rate and seasonality of precipitation, which depends on the regional climate setting of the drill site (Vinther et al., 2010, Zheng et al., 2018). Including other archives than ice cores would give a more widespread regional coverage, potentially providing better constraints on circulation patterns and climate trends. Some oxygen isotope records from tree rings in Sweden (Edwards et al., 2017) and speleothems from the European alps (De Jong et al., 2013) covering the past millennium primarily reflect winter climate conditions. Both records in these examples have 5-year resolution, and the speleothem record has hiatuses, which reflect some of the challenges in using these proxy records. However, there could be benefits of using a larger selection of data, despite the different temporal resolution (Steiger and Hakim, 2016)."*

3) The link with solar forcing is far from being straightforward, and the authors push their conclusions a bit too far I should say. Also, the variability is not necessarily related with external forcing, and large amount of variance can be purely stochastic (i.e. internal to the climate system). While lots of research is led on this topic, notably to explain

the little ice age, this is simply not discussed at all here. What is the percentage of variance in our reconstruction that is not related to any external forcing? This is not easy to isolate, but a rough estimate would be interesting (e.g. Wang et al. 2017).

*According to our statistical tests (taking into account autocorrelation on filtered data) the connection we find between solar forcing and the reconstruction is significant on a range of time scales (e.g. time lagged response in Table 2). We agree that variability in the climate system, and in particularly atmospheric variability, is partly governed by stochastic processes. However, we never claim that all of the variability is explained by the forcing, this is also not what the results show. The reviewer does not specify which studies we fail to discuss. We have tried to include additional relevant studies in the revision including Schurer et al. (2014) and Wang et al. (2017).*
*The study by Wang et al. (2017) is very different from our study as their reconstruction is represented by a single index of multi-decadal ocean variability, compared to our reconstruction, which is a climate field reconstruction of atmospheric variability. Their attributing of forced variability is done 30-yr low-pass filtered data for both forcings and reconstruction. Low-pass filtering of the volcanic forcing, comprised entirely of spikes of durations no longer than 3 years, alters the properties of the time series, creating an artificial forcing signal already before the eruptions happen due to the smoothing of the filter. This might be less of an issue when dealing with ocean indecies with slower variability. Given the abrupt atmospheric response to volcanic eruptions seen in our reconstruction we are reluctant to introduce this type of analysis without extensive testing of different methods, which would be a good topic for coming studies.*
*From our point of view the explained variability of forcings is already estimated in our analysis. For the 12 largest tropical volcanic eruptions we see an average of 5 years of NAO, which would correspond to 60 years out of 730 years in the reconstruction being affected by the forcing. In the case of solar forcing the correlation is up to 0.6 between PC2 of reconstructed SLP, corresponding to ~40% of explained variance at centennial time scales. As we are dealing with a reconstruction it is very uncertain how this explained variance translates to the real world, for example due to the explained variance of the reconstructed PCs having a different distribution compared to reanalysis data.*

With a proper discussion of the following points, I think the manuscript will clearly deserve to be published in climate of the past.

———————————— Specific points: ————————————
- p.1, l.1: the two first sentences can be interpreted as almost contradictory. Can you please reformulate them to clarify what you have in mind here?

*Corrected.*

- P.1, l. 3: add "of" before "the effects"

*Corrected.*

- P. 1, l. 3: "A positive phase. . .": This is not true. Some of the historical eruptions have been followed by negative phase of the NAO (e.g. Agung) and the link between NAO and volcanic eruptions over the historical era is very small and sensitive to the selected eruptions (cf. Swingedouw et al. 2017, cited in the ms.). Please clarify.

*Reformulated. This is the average response to tropical eruptions.*

- P. 1, l. 10: you should specify here that this is for winter season.

*Corrected.*

- P. 1, l. 16: "we observe a similar response" is not clear at all. Please reformulate what you mean here. The next sentence concerning "blocking frequency" is also quite unclear. Please avoid the word "likely" as it has a very specific meaning in IPCC report, which is not the one used here, since I have not seen any proper analysis of changes in blocking frequency in the manuscript.

*Reformulated, page 1, line 16-18:* *"On centennial time scales we observe a similar response in circulation as for the 5-year time-lagged response, with a high-pressure anomaly across North America and south of Greenland. This anomalous pressure pattern could be due to an increase in blocking frequency, possibly linked to a weakening of the subpolar gyre."*

- P. 1, l. 19: You'll need to define what you mean here by little ice age in terms of time period.

*Corrected. As the reviewer probably knows, there are some variations in how LIA is defined. Following others (e.g. Moffa-Sanchez et al., 2014) we define it 1450-1850 CE.*

- P. 1, l. 19: "a clear link" is quite subjective sentence. Can you be more specific (i.e. quantitative: the correlation is quite small so that the link is not that clear I would say).

*Reformulated. We now mention the range of correlation between solar forcing and reconstructed PC2 on decadal (r = 0.29, p < 0.01 (5-year time lag)) to centennial time scales (r = 0.6, p < 0.01).*

- P. 1, l. 20: no "s" to "pattern"

*Corrected.*

- P. 2, l. 1-2: It could be worth to specify here winter season, since the whole paper is focused on this season.

*Corrected.*

- P. 2, l. 6-7: "Past changes. . ..": Variability in atmospheric circulation is usually believed to be mainly stochastic, with a weak imprint of external forcing. This is why seasonal or decadal prediction are so difficult: the atmosphere is dominated by chaos, with very small part of the variance that is predictable. . . This is in contradiction with the word "attributed" from your sentence. Please clarify. Also, please add some references to substantiate the claim from this sentence.

*We have reformulated this sentence to clarify that we are discussing climate proxy studies that attribute variability in their records to external forcing, e.g. Wang et al. (2017).*

- P. 2, l. 10: add coma after "millennium"

*Corrected.*

- P. 2, l. 14-15: "The NAO has also. . .": This sentence is very confusing. Please rephrase by just saying that solar variations are slightly correlated with SLP over the historical era in a few areas.

*Reformulated. Many studies claim a NAO response to solar forcing, and it is common to refer to a NAO-like pattern. This is what we are discussing here.*

- P. 2, l. 23-24: what is also very important is the amplitude of the influence i.e. the variance explained by solar variability. It is possible this forcing has a slight influence, but how does it compare to the noise, i.e. the purely stochastic variation of the NAO. The signal-to-noise ratio is indeed key to evaluate here.

*We agree with his comment by the reviewer, which is also why we emphasize the preserved high frequency variability in our reconstruction. We assume that the reviewer is referring to line 22-23.*

*Inserted, page 2, line 15:* *"However, the apparent relationship between the NAO and solar forcing is not consistent during the 20th century (Gray et al., 2013)."*

- P. 3, l. 1: replace "particularly" by "particular".

*Corrected.*

- P. 3, l. 21-25: can you please provide a few more information on this model like its resolution? A brief description of its biases in the region analyzed will be also enlightening to evaluate the limit of this model for the exercise performed. How does the isotopic variations in the ice core sites compare with observations? How many grid points are covering Greenland?

*The atmospheric component of the model, ECHAM5, is a spectral model and we ran it in T31 resolution, corresponding to 3.75º x 3.75º (lon, lat), with 19 vertical layers. 5 of these layers are in the stratosphere. 50 grid points are covering Greenland. We have added this information on model resolution, and additional information on model performance in the methods section. Also, supplementary Figure S1, Table S1 and Table S2 is moved to the manuscript to help clarify the methods section. We now also discuss the range of isotope variability in model and reconstruction compared to ice core. We use stacked ice core data from DYE3 and GRIP to take into account postdepositional noise in the ice core data. We will show histograms in the revised supplementary (see figure below). Even stacking 5-6 ice cores does probably not take into account all noise in the ice core data, which is part of the motivation of not using stacked data and also matching the PCA loadings and not matching directly to the ice core data of individual cores.*

*Inserted, page 3, line 21-24:* *"The performance of the atmospheric component of the model used in this study, ECHAM5-wiso, was evaluated for the Arctic region and Antarctica using different configurations of spatial resolution by Werner et al. (2011). For the configuration used in this study (T31) the model has a warm bias and is not depleted enough in $\delta^{18}O$, how ever the climatological relation between d18O and temperature compares well to observations, despite the relatively course resolution."*

[Figure]

*New figure in revised supplementary. Histograms of ice core $\delta^{18}O$ and modeled $\delta^{18}O_{pw}$ for winter covering the period 1778-1970. **a** Histogram of stacked GRIP $\delta^{18}O$ (6 cores, gray), with $\delta^{18}O$ for a single core plotted in the background in light gray to illustrate reduction of variability from stacking. **b** modeled GRIP $\delta^{18}O$ (dark gray)with stacked ice core $\delta^{18}O$ plotted in the background for comparison. **c** same as **b**, but for reconstructed $\delta^{18}O$. **d-f**, same as **a-c**, but for DYE3. This figure both illustrates the inherent noise in ice core data, which can be reduced by using multiple cores, and shows that the model reconstruction covers most of the range of the ice core variability. Stacking even more ice core records (if available) could possible reduce the variability further. DYE3 has about twice the annual accumulation rate of GRIP, which also reduces the scatter.*

- P. 4, l. 11: this section 2.3 is key to the paper. Nevertheless, I find it a bit difficult to follow. They are very few references that support the method presented, so that it seems that this approach is entirely new. Is that correct? If so, I would like to have a longer description of it and a few examples to fully understand how the method works. Furthermore, I'm wondering how sensitive the results to the X2 measure are, which sounds a bit arbitrary and not very well justified.

*Our method is indeed entirely new. We have extended the description of important parts of the method. The Chi-square probability is a standard measure for the goodness of fit of a given model to an observation. It is eventually based on squared distances between model and observation (like the RMSE), but allows us to infer the probability for each ensemble member of not matching the variability of the ice core PCs. Admittedly, the cutoff at p > 0.01 is somewhat arbitrary, but it provides us with a reasonable number of ensemble members, and the p-value rapidly ramps up to ~1 at ensemble member 60. Hence, other distance measures may provide a different number of ensemble members, but as long as they are based on quadratic distances, they would identify the same ranking of model-data matches.*

- p. 5, l. 2: The authors depict "39 time series". This is not crystal clear what the time is here and what the members are. Can you please be more specific to improve this description (a scheme could be useful as well for instance).

*This should be 39 ensemble members, based on the best match of PC1, PC2 and PC3. We have clarified description of this section.*

*Reformulated, page 4, line 32 to page 5 line 1:* "We define the sorted model output as ensemble members, such that the best fitting model year, for each year of the ice core data, belongs to ensemble member 1, the second best fitting model years belong to ensemble member 2, and so forth. Using a Chi-square goodness-of-fit test, with respect to the measure in Eq. 1, we evaluate the ensemble members against the PCs of Greenland ice core $\delta^{18}O$ and reject model fits with likelihood $p > 0.01$ of not fitting the ice core data. This leaves us with 39 time series of reshuffled model data fitted to the ice core data."

- P. 5, l. 2: "reshuffled model": a model is a tool with which you can perform simulations, providing climate variables. Thus, I'm not sure that "reshuffled model" is a proper terminology. I think, you are speaking here of the "reshuffled output from the simulation". This remark is true throughout the manuscript where model is used inconsistently.

*The term we are using here is "reshuffled model fits" which could more accurately be "model output reshuffled to fit the ice core data". We have reformulated this sentence, and gone through the manuscript to correct for inconsistent terminology.*

- P. 5, l. 29: "Note. . .": This is not a very expression. Normally all what is written in the manuscript is worth to be noted.

*Corrected.*

- P. 5, l. 30: "is associated": it is not clear what is substantiating this claim. Please clarify.

*Reformulated. "... can be associated".*

-P. 5, l. 30: "Figure S3": I find the pattern of EOF2 very different between model simulation and reanalysis. This should be said somewhere.

*The difference in EOF patterns between model and 20CR is discussed page 6, line 4-10. We have clarified this discussion in the revision.*

*Inserted, page 6, line 9:* "This indicates that the variability projected on PC2 and PC3 of the reanalysis data is partly summarized in PC2 of the reconstruction."

- P. 5, l. 33: add a coma after "reconstructions"

*Corrected.*

- P. 6, l. 10: what about the biases of the model? I believe this can also explain your differences between model simulations and reanalysis! Climate models are not providing perfect representation of reality, the opposite is true.

*Here the expression "intrinsic model variability" includes model biases. We now discuss the role of the model run in the discussion. See also reply to major comment 1.*

- P. 6, l. 14 add "in" after "consists"

*We have added "of" after "consists".*

- P. 7, l. 16: "Scandinavian blocking-type pattern". You should support this by a reference. Also, I would have argued that a Scandinavian blocking is referring to a change in frequency and is usually an anticyclone that remains blocked over the Scandinavia, following the weather regime approach (e.g. Vautard et al. 1990, Ortega et al. 2014 in line with ice core analysis and weather signature). Can you please clarify this sentence?

*We have reformulated this sentence and added references (Ortega et al. 2014, Rimbu et al. 2016). Under the assumption that increased blocking frequency also results in higher average pressure in this region, we associate this pattern with the Scandinavian blocking pattern.*

- P. 7, l. 22: "it should be noted". Avoid this type of subjective comment.

*Corrected.*

- P. 7, l. 30: "looks slightly different": this is a very subjective judgments, and I would rather say that they have hardly anything in common. Can you please provide a more objective metric of their similarity (spatial correlation for instance)?

*As stated in line 28, we are comparing the 5-year time lagged response to the long-term response (Figure 4 b versus 4 c). We now refer to these figures for clarity.*

- P. 7, l. 31: "wave structure". OK, but the signs are almost opposite in Fig. 4 a and c. . . So, this is not very convincing as a similar pattern!

*See reply to comment above. The temperature response (Figure 4 e, and 4 f) is also very similar, and we now include this point in the comparison of the long-term and short-term response.*

- P. 8, l. 3: maybe state that this secondary pattern is usually denominated as the EAP. Also, it is better to avoid "likely", except when the meaning is in line with IPCC precise definition.

*We have reformulated this, also referring to the EAP pattern, and also add the range of correlations between PC2 and solar activity.*

- P. 8, l. 14: this positive feedback is very weak, and hardly significant in the observations. . . (which is why the NAO is so difficult to predict at the seasonal scale, and the recent improvements seem to be more related with tropical teleconnection and frontal dynamics around the Gulf Stream region. . .).

*For very strong volcanic eruptions, estimated to be well beyond the magnitude of eruptions during the instrumental era, forcing the NAO to a positive state for 2-3 years, the reinforcement feedback could play a bigger role. We have softened up this point in the revised discussion.*

*Reformulated, page 8, line 12-13: "... can best be explained ..." changed to "... could be explained ..."*

- P. 8, l. 30: this line is very speculative and not supported by any references. The influence of the North Atlantic SST on atmospheric circulation is mainly not significant

in winter season in the observation, or at least largely debated, so this hand-made explanation sounds a bit speculative I should say. I would at least replace "likely" l. 28 by "possible" given its high level of speculation.

*Corrected.*

- P. 9, l. 1: "Although the authors. . ." Indeed, they do not. . . They rather propose that the LIA would have been intrinsic and related to (unforced) rapid changes of the SPG. Furthermore, detection-attribution analysis (e.g. Schurer et al. 2014) found almost no signature of solar forcing on NH temperature, implying that LIA was hardy forced by solar variations. You should discuss this as well I would say. How do you reconcile this with your interpretations?

*The study by Moreno-Chamarro et al., shows the connection between the SPG and the anomalies in SLP. The weakening of the SPG might happen unforced in their model (the test of solar forcing is not specified), but we see good indications that the changes in atmospheric circulation (PC2) is coinciding with changes in solar activity on a range of time scales. The model study by Schurer et al. (2014) was, as mentioned by the reviewer, done on NH mean temperature. If we would average the North Atlantic response to solar forcing in reconstructed temperature we would likely also see no significant response as some areas get warmer, while others get colder. We have addressed these points in the revised discussion.*

*Inserted, page 9, line 3:* *"One explanation could be that low solar activity is the preconditioning factor in reality, causing the response to solar forcing seen in our reconstruction, while the climate response to solar forcing might not be fully captured by the MPI-ESM (Mitchell et al., 2015) used by Moreno-Chamarro et al. (2017)."*

*Inserted, page 9, line 4:* *"The complexity is also reflected in a non-uniform temperature response to solar forcing, with both regional warming and cooling. This also means that part of this signal will be smoothed out if such analysis is carried out on hemispherical mean temperature (e.g. Schurer et al., 2014)."*

- Figure 1: what about adding a sliding window correlation here? It could also be useful for Figure 2 and Figure 6. It will help to evaluate when the time series are well-correlated by another mean than just by eye.

*Following the suggestion of the reviewer we have added moving window correlations to the Figure 2 and 6, and the information from these new plots are included in the text (see plots below). In Figure 2 we used a 31-year window to show the variable strength in correlation, while we chose a 61-year window in Figure 6 to get reasonably clear correlations (~60 years cover ~5 solar cycles).*

[Figure]

*Revised Figure 2. Comparison of instrumental and proxy-based NAO reconstructions. **a** Moving 31-point correlation between reconstructed NAO from this study and NAOcc (magenta), NAOmc (green) (Ortega et al., 2015) and observed NAO (yellow) (Jones et al., 1997). Only significant correlations are plotted (p < 0.05). **b** Ensemble mean reconstructed NAO (PC1 of reconstructed SLP (Hurrell et al., 2013)) with error estimated by ensemble spread and RMSE, compared to observed NAO (Jones et al., 1997) and NAO reconstructions by Ortega et al. (2015). The amplitude of all time series are scaled to fit the decadal variability of the observed NAO. **c** Same as **b**, except filtered with a 30 point 'loess' filter.*

[Figure]

*Revised Figure 6. Reconstructed PC2 of SLP plotted with the 5-year lagged sunspot number.* **a** *Moving 61-point correlation between reconstructed PC2 of SLP and sunspot number shifted for a 5-year time lag.* **b** *Time series of sunspot number shifted for a 5-year time lag, PC2+PC3 of 20CR SLP (see text and Table 2) and PC2 of reconstructed SLP.* **c** *Same as* **b**, *except filtered with a 5-year low-pass filter.* **d** *Same as* **b**, *except filtered with a 20-year low-pass filter.*

*References*

*de Jong, R., C. Kamenik, M. Grosjean Cold-season temperatures in the European Alps during the past millennium: variability, seasonality and recent trends, Quat. Sci. Rev., 82 (2013), pp. 1-12*

*Edwards, T. W. et al. Seasonal variability in Northern Hemisphere atmospheric circulation during the Medieval Climate Anomaly and the Little Ice Age. Quat. Sci. Rev. **165**, 102–110 (2017).*

*Masson-Delmotte, V., et al. (2008), A review of Antarctic surface snow isotopic composition: Observations, atmospheric circulation, and isotopic modeling, J. Clim., 21(13), 3359–3387, doi:10.1175/2007JCLI2139.1.*

*Mitchell, D. M., Misios, S. , Gray, L. J., Tourpali, K. , Matthes, K. , Hood, L. , Schmidt, H. , Chiodo, G. , Thiéblemont, R. , Rozanov, E. , Shindell, D. and Krivolutsky, A. (2015), Solar signals in CMIP-*

5 simulations: the stratospheric pathway. Q.J.R. Meteorol. Soc., 141: 2390-2403. doi:10.1002/qj.2530

Ortega, P., Swingedouw, D., Masson-Delmotte, V., Risi, C., Vinther, B., Yiou, P., Vautard, R., Yoshimura, K., 2014. Characterizing atmospheric circulation signals in. Greenland ice cores: insights from a weather regime approach. Clim. Dyn. 43 (9–10), 2585–2605.

Rimbu, N., Lohmann, G., Werner, M., & Ionita, M. (2017). Links between central Greenland stable isotopes, blocking and extreme climate variability over Europe at decadal to multidecadal time scales. Climate Dynamics, 49, 649–663.

Roeckner, E., Bäuml, G., Bonaventura, L., Brokopf, R., Esch, M., G., Hagemann, S., Kirchner, I., Kornblueh, L., Manzini, E., Rhodin, A., Schlese, U., Schulzweida, U., and Tompkins, A.: The atmospheric general circulation model ECHAM5. Part I: Model description., Tech. Rep. Rep. 349, 127 pp., Max Planck Institute for Meteorology, available from MPI for Meteorology, Bundesstr. 53, 20146 Hamburg, Germany, 2003.

Schurer, A. P., Tett, S. F. B. & Hegerl, G. C. Small influence of solar variability on climate over the past millennium. Nature Geosci. 7, 104–108 (2014).

Sjolte, J., G. Hoffmann, S. J. Johnsen, B. M. Vinther, V. Masson-Delmotte, and C. Sturm (2011), Modeling the water isotopes in Greenland precipitation 1959–2001 with the meso-scale model REMO-iso, J. Geophys. Res., 116, D18105, doi:10.1029/2010JD015287.

Steiger, N. and Hakim, G.: Multi-timescale data assimilation for atmosphere–ocean state estimates, Clim. Past, 12, 1375-1388, https://doi.org/10.5194/cp-12-1375-2016, 2016.

Zheng, M., Sjolte, J., Adolphi, F., Vinther, B. M., Steen-Larsen, H. C., Popp, T. J., and Muscheler, R.: Climate information preserved in seasonal water isotope at NEEM: relationships with temperature, circulation and sea ice, Clim. Past Discuss., https://doi.org/10.5194/cp-2018-8, in review, 2018.

Wang, J., Yang, B., Ljungqvist, F. C., Luterbacher, J., Osborn, T. J., Briffa, K. R., and Zorita, E.: Internal and external forcing of multidecadal Atlantic climate variability over the past 1,200 years, Nat. Geosci., 10, 512–517, 2017.

Werner, M., P. M. Langebroek, T. Carlsen, M. Herold, and G. Lohmann (2011), Stable water isotopes in the ECHAM5 general circulation model: Toward high-resolution isotope modeling on a global scale, J. Geophys. Res., 116, D15109, doi:10.1029/2011JD015681.

---

## Author Response (AR2)

Dear Marit-Solveig,

Thank you for the careful reading of our manuscript. We have revised the manuscript according to you comments. Please find our reply to your comments below *in italics*. We have attached a revised manuscript with tracked changes after our response.
* * *
In the abstract you write: "While our results show significant correlation between solar forcing and the secondary circulation pattern on decadal (r = 0.29, p <0.01) and centennial timescales (r= 0.6, p < 0.01), we find no consistent relationship between solar forcing and NAO."
Your conclusion states: "Using this approach, tropical volcanic forcing accounts for about 10% of the decadal to multi-decadal variability of the reconstructed NAO, while solar forcing accounts for about 40% of the variability of PC2 of reconstructed SLP on centennial timescales."

In the caption of revised figure 2 you write: "Ensemble mean reconstructed NAO (PC1 of reconstructed SLP (Hurrell et al., 2013))…"

In the first couple of reads it made me somewhat confused, as SLP is after all used an an index of NAO and it was only when I finally noticed the difference in PC number and your much clearer statement in the introduction that I understood the conclusions: "We find the average response to major tropical volcanic eruptions to be a positive NAO for the five consecutive winters after eruptions, which is more persistent 20 than previous studies have shown. However, we find no persistent relationship between solar forcing and the NAO. On the other hand, we find a strong impact of solar forcing on the secondary modes of circulation represented by the second principal component (PC2) of reconstructed sea level pressure (SLP).We achieve the strongest correspondence between the solar forcing and reconstructed PC2 of SLP with a time lag of 5 years, indicating that an atmosphere-ocean feedback is in play. Taking this time lag into account we find a consistent relationship between PC2 of 25 reconstructed SLP and solar forcing on decadal to centennial time scales".

I would therefore ask you to make this also clearer in the conclusions and abstract. Also, at one point you mention that PC1 of SLP = NAO, while PC2 of SLP may correspond to the East Atlantic pattern. Could you expand on this, if possibly also refer to this in the conclusions?

> *We agree that this important point of the paper could have been written in a clearer way. We now define the NAO and the Eastern Atlantic pattern at the beginning of the introduction, also explaining the definition of the NAO using station data and using principal component analysis. The definition of the NAO (PC1) and the Eastern Atlantic pattern (PC2) is now used throughout the abstract and conclusions, to make this point more clear.*

[revised manuscript text omitted]

---

## Author Response (AR3)

Dear Editorial office,

I have uploaded a new revision of the manuscript where is only change is adding a section on data availability. The data connected to the publication has been submitted to the PANGAEA open access library, however, the submission of the data will not be complete until two to three weeks from now. So, presently the is no DOI available for this data submission.

Best wishes,

Jesper Sjolte